# Lipidic folding pathway of α-Synuclein via a toxic oligomer

Vrinda Sant[1], Dirk Matthes[2], Hisham Mazal [3,4], Leif Antonschmidt [1], Franz Wieser[3,4,5], Kumar T. Movellan [1,8], Kai Xue [1,9], Evgeny Nimerovsky[1], Marianna Stampolaki [1], Magdeline Nathan[1], Dietmar Riedel[6], Stefan Becker [1], Vahid Sandoghdar [3,4,5], Bert L. de Groot [2] ✉, Christian Griesinger [1,7] ✉ & Loren B. Andreas [1] ✉

Aggregation intermediates play a pivotal role in the assembly of amyloid fibrils, which are central to the pathogenesis of neurodegenerative diseases. The structures of filamentous intermediates and mature fibrils are now efficiently determined by single-particle cryo-electron microscopy. By contrast, smaller pre-fibrillar α-Synuclein (αS) oligomers, crucial for initiating amyloidogenesis, remain largely uncharacterized. We report an atomic-resolution structural characterization of a toxic pre-fibrillar aggregation intermediate (I1) on pathway to the formation of lipidic fibrils, which incorporate lipid molecules on protofilament surfaces during fibril growth on membranes. Super-resolution microscopy reveals a tetrameric state, providing insights into the early oligomeric assembly. Time resolved nuclear magnetic resonance (NMR) measurements uncover a structural reorganization essential for the transition of I1 to mature lipidic L2 fibrils. The reorganization involves the transformation of anti-parallel β-strands during the pre-fibrillar I1 state into a β-arc characteristic of amyloid fibrils. This structural reconfiguration occurs in a conserved structural kernel shared by a vast number of αS-fibril polymorphs including extracted fibrils from Parkinson's and Lewy Body Dementia patients. Consistent with reports of anti-parallel β-strands being a defining feature of toxic αS pre-fibrillar intermediates, I1 impacts viability of neuroblasts and disrupts cell membranes, resulting in an increased calcium influx. Our results integrate the occurrence of anti-parallel β-strands as salient features of toxic oligomers with their significant role in the amyloid fibril assembly pathway. These structural insights have implications for the development of therapies and biomarkers.

The aberrant aggregation of α-Synuclein (αS) into amyloid fibrils is a crucial step in the biochemical cascade of several neurodegenerative diseases (NDD) as evidenced by the fact that αS amyloid fibrils are a major component of Lewy bodies, the intra-cellular inclusions that are characteristic of Parkinson's disease and other synucleinopathies[1,2]. While fibrils are a pathological hallmark of NDDs, evidence has accumulated that oligomeric αS aggregation intermediates, in particular, exert a toxic load on neurons[3–6]. Further, the ability of αS to interact with and disrupt lipid bilayers is well documented[3,7,8]. Hence, such structures in complex with lipids, a canonical binding partner of αS[9–12], are of particular interest.

**Fig. 1 | Comparison of I1 oligomer and L2 fibril. A** Transmission electron micrograph (TEM) image of I1 aggregates (left) compared to a TEM image of L2 fibrils (right) superimposed on an aggregation kinetics curve (black curve). Scale bar 100 nm. The ThT kinetics curve is a schematic representation. Statistics are reported in Supplementary Fig. 1E. Insets show schematics of structures and highlight unknown aspects of I1. The gray curve shows slower aggregation kinetics under conditions during NMR measurements (4–16 °C and monomer depleted). **B** Backbone traces of αS fibril polymorphs that have a β-arc at T59 similar to the L2 fibrils. **C** Impact of αS aggregates on viability of SH-SY5Y neuroblastoma cells after 24 h of incubation with 0.3 μM (gray bars) or 0.6 μM (black bars) αS. Error bars are shown as the ± standard error of the mean for 6 replicates. * indicates $p < 0.05$ and ** indicates $p < 0.01$, from a one-way ANOVA Tukey test. The lipid concentrations corresponding to each sample are shown in parentheses on the x-axis. Lipid concentration is shown in gray or black corresponding to 0.3 μM αS or 0.6 μM αS, respectively. Circles show individual datapoints. **D** The secondary structure as indicated by chemical shift, is shown as helix (waves), strand (arrows) or loops (lines) for Intermediate 1 (I1) and the L2 fibril. Chemical shift similarity (green-pink) mapped onto the sequence of αS. Dotted lines represent tentative assignments. White spaces indicate missing assignments. Gray lines denote assigned residues of I1 that are unassigned in the L2 fibril. **E** Per residue average chemical shift perturbations (CSPs), including Cα, Cβ, Co and N$_H$ shifts, between I1 and L2 fibril (BMRB 50585). Dotted line shows the 0.7 ppm cut-off for similar and dissimilar segments. Source data are provided as a Source Data file. **D**, **E** Residues with similar chemical shifts (CSPs <0.7 ppm) are colored green. Residues with dissimilar chemical shifts (CSP > 0.7 ppm) are pink. Similarity for the helical segment V16-T22 is derived from ${}^{13}$C correlation spectra only[17]. **F** Side-chain contacts observed for I1 (from (H)HNH and (H)HCH spectra in Supplementary Fig. 3B) conflicting with the L2 fibril structure (grey trace) are marked in pink.

In vitro preparations have been instrumental in determining characteristics of intermediate species occurring during amyloid aggregation because their low population and transient nature make it challenging to isolate them from tissues. Through these studies it has been revealed that aggregation intermediates, sometimes transient fibril like filaments[13], are often composed of segments with structural features analogous to their respective fibrillar polymorphs[14–17]. However, atomic-resolution structures of small oligomeric αS intermediates are lacking. This impedes the understanding of nucleation, toxicity, and the effect of aggregation modulators at the molecular level.

Previously, we reported the isolation of Intermediate 1 (I1), a transient pre-fibrillar species found on pathway to the formation of the L2 fibril polymorph (Protein Data Bank (PDB) entry 8A4L in the presence of anionic lipid vesicles composed of a 1:1 molar ratio of POPA and POPC[17,18] (Fig. 1A). Nuclear Magnetic Resonance (NMR) chemical shifts indicated that I1 shares several segments with the L2 fibril, including residues L38-S42 in β1, T44-V48 in loop 1, E57-E61 in loop 2, T72-A78 in loop 3 and β4[17] (Fig. 1D, E).

A β-arc, which is a characteristic feature of amyloid fibrils, is found at T59 (V52-V66) in the L2 fibril. This β-arc is a structural kernel conserved in nearly half of the deposited αS fibril polymorphs (Fig. 1B), including extracted fibrils from Parkinson's disease (PD) and Lewy

Body Dementia (DLB) patients (PDB 8A9L) and those seeded from Multiple System Atrophy (MSA) (PDB 7NCA) and PD patients (PDB 7OZG). Determination of the structure of I1, which clearly differs from the fibril for residues V52-V66 (Fig. 1A, D, E)[17], would help elucidate the folding pathway for L2, thus far unknown[7,13].

Here, we report extensive NMR data for I1 that reveal an antiparallel β-sheet with a β-hairpin at T59. Together with super-resolution fluorescence microscopy data, I1 was determined to be a tetramer. Atomistic molecular dynamics (MD) simulations reveal that the tetramer is stabilized in the context of a lipid bilayer.

## Results

### Composition and stability of an I1 sample

The characterization of an I1 sample, previously isolated on the folding pathway to the L2 fibril[17], reveals its prolonged stability and composition. The I1 sample can be isolated for prolonged times in the rotor (several weeks), which we attribute to a reduction in temperature from 37 °C during aggregation to below about 20 °C during NMR measurements (Fig. 1A). Additionally, stability might be improved because I1 has been depleted in disordered monomer and membrane bound monomer via ultracentrifugation before packing. Fingerprint spectra are acquired at regular intervals to keep track of the stability of I1

(Supplementary Fig. 1A). Lipid bound monomer and disordered monomer stay in the supernatant after I1 is isolated (Supplementary Fig. 1B, C). Due to its transient nature, multiple freshly prepared samples of I1 have been used in the study (Supplementary Fig. 1E). The spectra indicate that I1 consists of one dominant species (Supplementary Fig. 1 and 2). Additionally, there was no indication of the L2 fibril in the I1 samples, as determined by the absence of characteristic L2 resonances in the I1 spectra (Supplementary Fig. 1). Notably, no instance occurred where one residue was assigned to two sets of resonances (Supplementary Fig. 2).

### I1 is toxic to SH-SY5Y neuroblasts

Consistent with well-established behavior of amyloid aggregates[6,8], I1 and the L2 fibril dramatically differ in their impact on cell viability: I1 reduces survival of SH-SY5Y neuroblastoma cells to 25%, while the L2 fibril leaves cell viability unperturbed compared to the lipid bound monomer (Fig. 1C). This is despite the remarkable similarity between the two species and further motivates a detailed characterization to link structural differences to variations in cellular impacts.

### Distinct features of I1 compared to the L2 fibril

Clear differences are seen between I1 and the L2 fibril in morphology as well as secondary structure and topology. An I1 sample resolubilized from an MAS NMR rotor consists of particles of diameter 8–15 nm, drastically different from longitudinal, twisted strands observed for fibrils (Fig. 1A). Comparison of C$\alpha$ and C$\beta$ chemical shifts (BMRB entries 50585[17] and 52283) reveals major differences between I1 and L2 in two regions of $\alpha$S. Firstly, C-terminal residues, E83-K97, that are primarily structured as a $\beta$-strand in the L2 fibril, form a loop and $\alpha$-helix in I1 (Fig. 1D). Differences in topology are also observed in this segment: K96 shows contacts to residues around A30 in I1. By contrast, residues V82-L97 are adjacent to $\beta$3 in the L2 fibril (Supplementary Fig. 3B, C). Secondly, the $\beta$2 and $\beta$3 strands, while retaining $\beta$-strand secondary structure in both, I1 and the L2 fibril (Fig. 1D), deviate substantially in their chemical shifts (average CSP > 0.7 ppm) (Fig. 1E). These segments exhibit side-chain contacts in I1, that conflict with the L2 fibril, namely N65 N$_\delta$ – A53 H$_N$ and V63 H$_\gamma$ – T54 H$_\beta$, (Fig. 1F). In the L2 fibril these contacts measure greater than 10 Å, which is beyond the distance reached in H(H)NH NMR spectra. In the L2 fibril, $\beta$2 and $\beta$3 form the T59 $\beta$-arc, a shared structural kernel among various fibril polymorphs, suggesting possible commonalities in the folding pathway of other fibrils sharing the T59 $\beta$-arc.

To determine the $\beta$-strand arrangement in I1, we recorded amide proton correlation spectra[19] on the 1.2 GHz spectrometer to leverage improved resolution and sensitivity (Supplementary Fig. 3A). Parallel-in-register (PIR) and anti-parallel (AP) $\beta$-strands produce distinct contact maps of proximity among amide moieties. $\beta$1 and $\beta$4 are confirmed as PIR, since only correlations to neighboring residues are observed (green labels, for example K43-T44 in Fig. 2A). In contrast, the pattern of amide proton correlations for $\beta$2 and $\beta$3 reveals an AP arrangement (pink labels in Fig. 2A). These correlations for V52-V55 on $\beta$2 with V66-V63 on $\beta$3 are depicted in Fig. 2B. To distinguish intra- and intermolecular amide proton contacts, we diluted the uniformly $^{15}$N-labelled $\alpha$S with 50% unlabelled $\alpha$S. Once normalized by the diagonal, the intensities of intermolecular contacts in the diluted labelling spectrum (blue, Fig. 2A) are reduced 2-fold compared to the fully labelled spectrum (black, Fig. 2A), while intramolecular contacts retain full intensity. In this way we identified the V66-V52 (AP) and K43-T44 (PIR) cross-peaks as intermolecular whereas the V63-V55 cross-peak was identified as intramolecular (Fig. 2C).

### Oligomer state of I1

Amide proton correlations indicate that I1 is a multimer, prompting further investigation with NMR and fluorescence measurements to determine oligomer size. An NMR CODEX[20] (Center band only detection of exchange) measurement allows for spin counting with an upper limit of about 10 Å. When each molecule is labeled at a single site, CODEX can be used to determine the oligomer number, provided that the labeled sites form a cluster with the nearest intra-spin distance below 10 Å. For these measurements, I1 was prepared with $\alpha$S containing a single $^{13}$C isotopically labeled site at H50 C$\epsilon$. A CODEX measurement involves the decay of initial magnetization of this single isotope labeled nucleus until the signal plateaus at the inverse of the number of spins over which magnetization can equilibrate. The CODEX curve reaches about 0.25 at long times, indicating that I1 is at least a 4-mer (Supplementary Fig. 4A).

Stepwise photobleaching can be used to count the number of monomers in an aggregate[21]. In this work, we combine this approach with polarCOLD, a cryogenic super-resolution fluorescence microscopy, which can reach Ångstrom resolution[22,23]. We first verified that modification of a portion of molecules in I1 with a fluorophore showed no perturbations in NMR structural data and aggregation kinetics (Supplementary Fig. 4C, E). Next, aggregates were immobilized on a substrate and irradiated continuously at room temperature (RT) until they photobleached (Supplementary Fig. 4F). For these measurements, I1 was prepared by diluting dye labeled $\alpha$S with wild-type unlabeled $\alpha$S at a 3:1 ratio. Stochastic mixing of dye labeled and unlabeled $\alpha$S molecules results in aggregates with varying numbers of dye-labeled molecules, leading to some aggregates being fully labeled, some with no labels and others with partial labeling. This distribution of dye molecules results in a corresponding distribution in the number of photobleaching events, which follows a binomial distribution. The number of photobleaching steps was determined by counting the intensity levels in the time-traces (Supplementary Fig. 4G). The histogram of photobleaching steps at RT can be best fit to a binomial distribution for a tetramer (purple, Fig. 2D). Furthermore, polarCOLD[23,24], was used to acquire and quantify super-resolution images at a temperature of 4 K by localizing individual fluorophores through their emission polarization states. Examples of such images for the projection of fluorophore positions are depicted in Fig. 2E–H for different particles from a single preparation of I1 with ~27% dye labeling. We also analyzed an ensemble of individual particles to obtain a histogram of polarization states (Supplementary Fig. 4H) per aggregate at 4 K (pink in Fig. 2D), yielding very good agreement with the RT measurements. These measurements all indicate that I1 is a tetramer.

### Visualization of the I1 tetramer structure

An atomic resolution model for an I1 tetramer was assembled by combining the knowledge of the L2 fibril structure (for I1 segments similar to the fibril) with experimental contacts observed for I1 that are distinct from the L2 fibril (Supplementary Table 1). Detection of a single set of chemical shifts suggests a single fold for all monomers of I1, but several quaternary arrangements can be modeled to satisfy experimental restraints (Supplementary Fig. 5). These include "open" arrangements where molecules simply stack on each other like in the fibril (Fig. 3A). We can also envisage "closed" arrangements, such as a "barrel" with inter-molecular H-bonds for all four molecules or a "bowl" morphology with intra-molecular H-bonds for all molecules except one (Supplementary Fig. 5A). A comparison of the conformers reveals two distinct structural features in all morphologies (Fig. 3B, C, Supplementary Fig. 5A). The first is a fibril-like PIR arrangement of $\beta$1 and $\beta$4 (green, Fig. 3). The second is an AP domain that involves a $\beta$-hairpin between strands $\beta$2 and $\beta$3 connected by loop 2 (pink, Fig. 3). The AP domain in both open and closed models satisfies the 7 Å upper-limit for side-chain contacts in I1 (Fig. 3D).

### Transition from $\beta$-hairpin in I1 to $\beta$-arc is necessary for formation of the L2 fibril

A $\beta$-hairpin turn is characterized by backbone hydrogen bonds between consecutive $\beta$-strands, as seen in I1. On the other hand, a

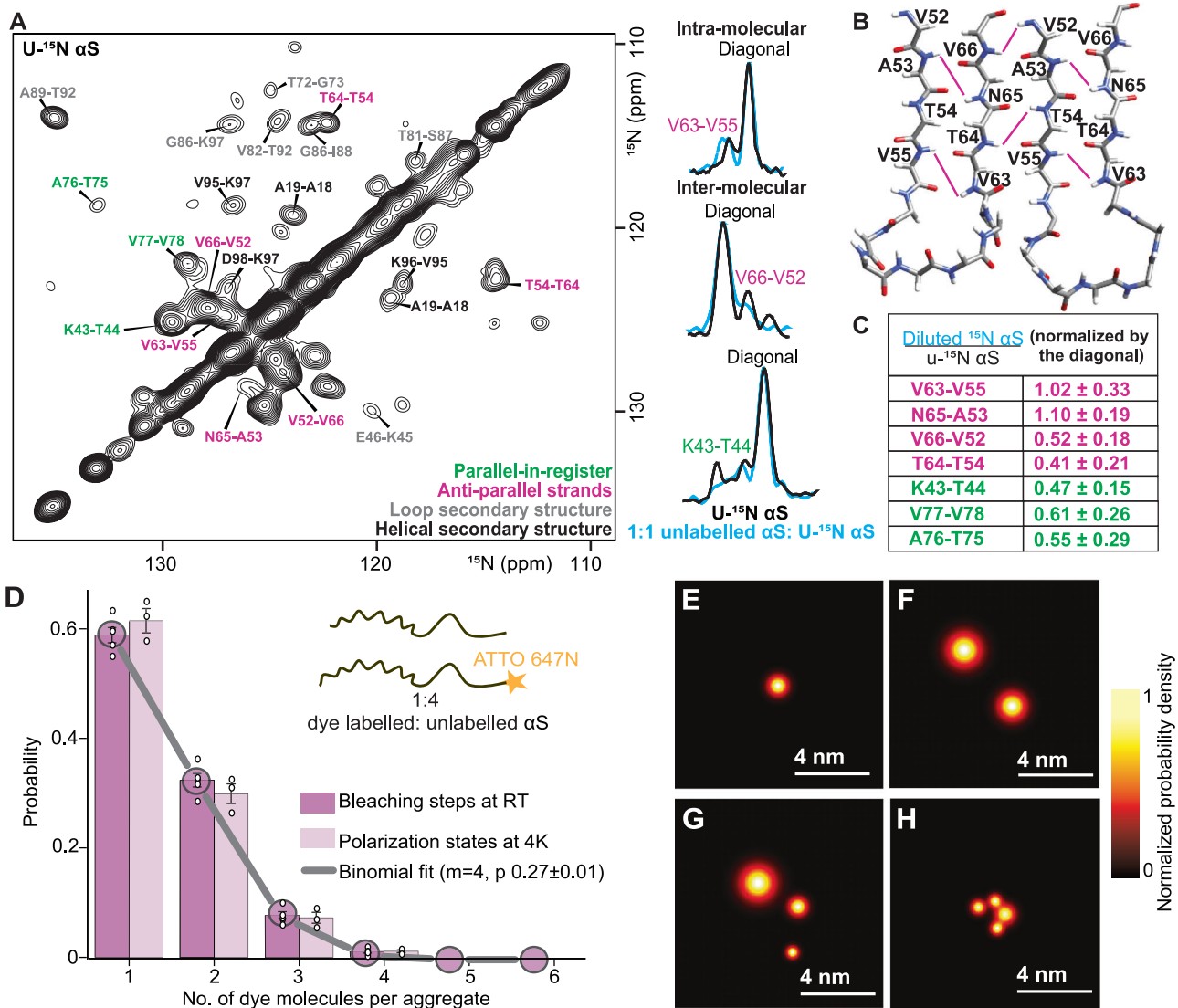

**Fig. 2 | I1 has inter- and intra- molecular (AP) β-strands and is a tetramer.**
**A** $^{15}$N-$^{15}$N correlation spectrum of I1 with uniformly $^{15}$N- labelled αS shows that there are AP segments (pink labels) which feature intra- and inter-molecular cross-peaks. Cross-peaks from PIR segments (green labels), correlations in loops (grey labels) and helices (black labels) are also observed. On the right are one-dimensional traces of cross-peaks comparing uniformly $^{15}$N- labelled I1 (black) and 1:1 $^{15}$N: unlabelled (50% dilute) I1 (blue). Intra-molecular cross-peaks are not affected by the 50% dilution (V63-V55 contact) whereas the intensity of inter-molecular cross-peaks is reduced 2-fold in the diluted spectrum (blue) (V66-V52 and K45-T44 contacts). **B** Contacts indicative of AP segments are indicated with pink lines on a backbone trace. **C** Ratios of cross-peak intensity between the fully $^{15}$N-labeled (black) and 50% $^{15}$N-labeled (blue) spectra. Color legend is the same as in panel (A). **D** Histograms of the number of photobleaching steps (purple) and number of polarization states per aggregate (pink) for I1 at a fluorophore labelling efficiency ($p$) of 25%. The grey line

shows the best fit to the photobleaching and polarCOLD data with parameters $p = 27\%$ and $m$ (no. of monomers per oligomer) =4 (Supplementary Fig. 4I). Error bars represent standard error of the mean calculated from 1421 particles (over 5 different fields of view (FOV)) for the photobleaching experiment and 1023 particles (over 3 different FOV) for the polarCOLD experiment. Probabilities for individual FOVs are shown in circles. Source data are provided as a Source Data file. **E**–**H** Examples of super-resolved polarCOLD images of different particles show the projection of aggregates with 1 (E), 2 (F), 3 (G) and 4 (H) dye molecules onto the imaging plane. The center of each spot displays the position of the fluorophore and its width represents the localization precision. The latter can vary due to the available signal-to-noise in each case, determined by the photophysics heterogeneity of fluorophores (Supplementary Fig. 4H). The number of particles with states in panel E-F can be determined from the polarCOLD histogram in panel D and the source data provided in the Source Data file.

β-arc is characteristic of amyloid fibrils, and displays a distinct backbone hydrogen bonding pattern from β-hairpin turns (Supplementary Fig. 6A, left). A β-arc is formed when two consecutive β-strands interact via their side-chains instead of the backbone. In such a scenario, the backbone hydrogen bonds are perpendicular to the β-arc and in fibrils are observed to form between two molecules, resulting in stacking of these molecules (Supplementary Fig. 6A, right).

To convert to the L2 fibril, I1 must undergo a transition in the AP domain from a β-hairpin, with backbone H-bonds between β2 and β3, to a β-arc with side-chain interactions between β2 and β3 instead. The

kinetic stability of I1 is partly attributed to the relatively high energy barrier required for breaking 28 H-bonds involved in this transition (Supplementary Movie 1). Contrary to an amyloid fibril, an I1 type conformation cannot template an indefinite number of molecules. Every additional molecule that contributes one layer to the PIR domain, adds two layers to the AP domain. This causes frustration between the two domains in larger aggregates and manifests as steric clashes in the G67-V74 segment and discontinuities in AP β-strands (Supplementary Fig. 5B), making it energetically unfavorable to template both the AP and PIR domain onto additional molecules (Supplementary Fig. 5C).

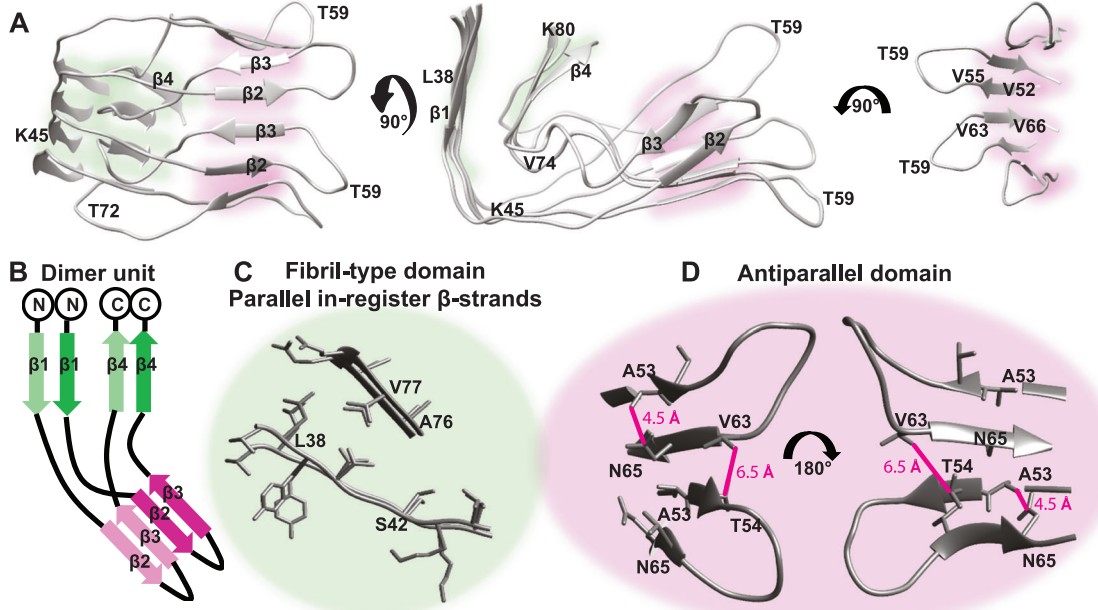

**Fig. 3 | I1 fold contains AP β-structure and PIR strands. A** Tetramer with open AP β-strands 2 and 3 and **B** Schematic of a dimeric structural element of the tetramers. The fibril-like PIR domain is colored green. The AP domain is colored pink. **C** Close up of the PIR part. **D** Close up of the AP arrangement showing the contacts N65 $N_\delta$–A53 $H_N$ and V63 $H_\gamma$–T54 $H_\beta$ as pink lines. The structure satisfies an upper limit of 7 Å.

This is consistent with the finding that Intermediate 2 (I2), the next intermediate on the pathway, features a β-arc in the V52-V66 segment, and thus a conversion from AP to PIR β-strands in this region, indicated by similar chemical shifts to the L2 fibril[17] and next neighbor correlations in the amide proton correlation spectrum (Supplementary Fig. 6B). Characteristic of fibrillar intermediates, I2 exhibits filamentous morphology (Supplementary Fig. 6C) and coincides with a rapid increase in ThT (ThioflavinT) fluorescence (Supplementary Fig. 6D) indicating fibril growth through the necessary transition from β-hairpin between the AP β-strands in I1 to β-arc between the parallel β-strands.

### Interaction of I1 with lipids

I1 has lipid interactions spanning the entire length of the protein, consistent with its aggregation on POPC/POPA vesicles[17], (Fig. 4A, B and Supplementary Fig. 7). This includes the N-terminal helix and hydrophobic residues 70–88 (Fig. 4B, Supplementary Fig. 7) consistent with previously observed αS oligomers[3]. Additionally, I1 also contacts lipids through Y39 and through the AP domain. The lipid contacts at Y39 are similar in the L2 fibril and I1, while L2 type lipid contacts at β4 are missing in I1.

All-atom MD simulations were performed to probe interactions of I1 with lipid bilayers. Simulations of several orientations of I1 with respect to bilayers with a truncated G36-T81 segment identified orientations compatible with experimental lipid contacts (Supplementary Note 1, Supplementary Fig. 8). Simulations that include the helical regions (V16-Q99), predict that the open morphology in orientations 1 and 2 and the bowl morphology in orientation 2 satisfy experimentally observed lipid contacts and distance restraints, as well as secondary structure propensity, with high fidelity (Fig. 4C, D, Supplementary Fig. 9 and Supplementary Fig. 10C, E, F).

Lipids appear to play a crucial role in stabilizing the AP domain. The E57-V66 segment in the AP domain contacts lipids in I1 (Fig. 4B), while in the L2 fibril, this segment forms a PIR β-sheet and does not show any lipid contacts. Notably, MD simulations of I1 with the AP domain oriented outside the lipid bilayer do not agree with experimental restraints and show a significant loss in β-strand content

(orientations 3, and 5-8 in Supplementary Fig. 8 and orientation 3 in Supplementary Fig. 9).

### I1 disrupts membranes

Strikingly, a cluster of charged residues in loop 1 (43KTKE46) and the AP domain (57EKTKEQ62) of I1 has contacts with lipid acyl chains (Fig. 4B). Simulations of the open tetramer show that lipid bending stabilizes these residues (Fig. 4C, D and Supplementary Movie 2 and 3), reducing the energy barrier for water and the penetration of choline and phosphate groups of the lipid headgroups into the bilayers hydrophobic core (Supplementary Fig. 10B, D). This polar defect created by the stable and membrane-inserted AP domain of I1 (Supplementary Fig. 10G) also facilitates cation flux across the membrane by lowering the energy barrier for $Na^+$ and $Ca^{2+}$ and ions in the hydrophobic bilayer core in MD simulations.

The disruptive impact of I1 on lipid membranes is evident from a liposomal proton flux assay. After establishing a pH gradient across the membrane by the addition of an acid, I1 triggers a proton influx into liposomes, gradually increasing the pH of the external buffer (Fig. 4E). By contrast, in the absence of I1, liposomes are sealed and maintain a stable pH. Furthermore, I1 increases $Ca^{2+}$ influx across neuroblast cell membranes, as indicated by the enhanced fluorescence of the calcium sensitive dye, Fluo-4 (Fig. 4F). Propidium Iodide fluorescence confirms that increased Fluo-4 fluorescence is not due to cell death induced membrane damage (Supplementary Fig. 11E). Unlike with I1, both monomeric and fibrillar αS show no significant differences in $Ca^{2+}$ influx (Supplementary Fig. 11F). The influx of $Ca^{2+}$ induced by I1 is independent of AMPAR channels, shown by the unchanged $Ca^{2+}$ influx curves with the inhibitor, cyanquixaline (CNQX) (Supplementary Fig. 11G).

### Discussion

Since there is no information about oligomers ex vivo, we compared the end product of our in vitro oligomer preparation, the L2 fibrils with the ex vivo Lewy fold. The latter closely resembles L2 but differs in notable aspects. The Lewy fold fibrils occur only as single filaments, whereas L2 consists of three filaments (Supplementary Fig. 12A). There

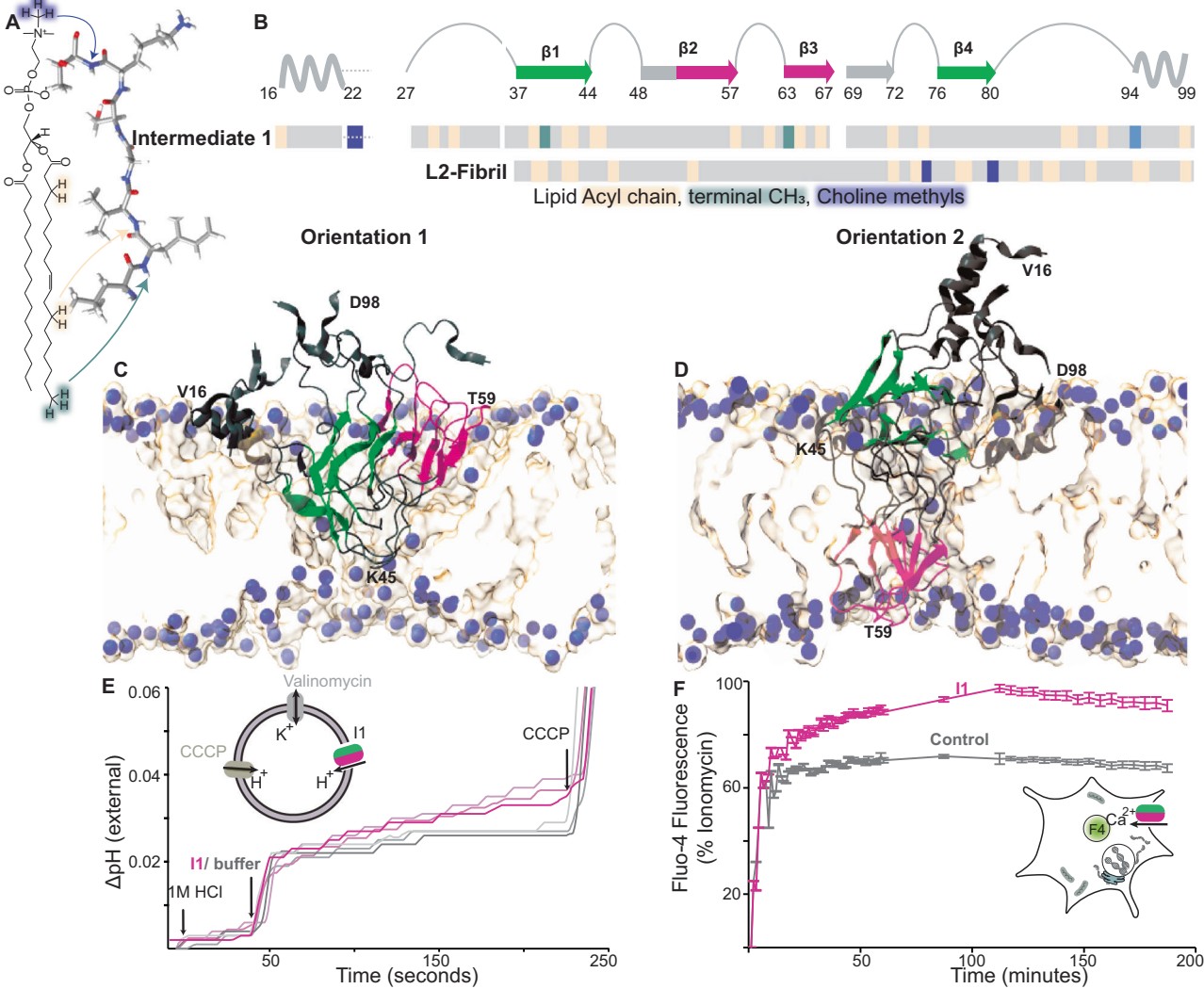

**Fig. 4 | PIR and AP domains of I1 are lipid bound and disrupt lipid membranes.**
**A** Schematic of magnetization transfer between lipid protons and protein back-bone. POPC choline nitrogen atoms are highlighted in purple, terminal methyl groups in the hydrophobic core are turquoise and acyl chain protons are light orange. **B** lipid contacts mapped onto the sequence for I1 and the L2 fibril[17]. Lipid contacts are colored according to lipid protons in panel (A) and are shown together with a schematic of the I1 secondary structure along its sequence. The PIR domain is colored green, and the AP domain is pink. Residues with missing assignments are shown as white spaces and those with tentative assignments are shown as dotted lines. Snapshots from unrestrained MD simulations of an open I1 conformer with (**C**) both PIR and AP domains being in the same leaflet and (**D**) PIR and AP domains traversing both leaflets of the bilayer. The bilayer in the simulation is composed of 1:1 molar ratio of POPC and POPA. POPC choline nitrogen atoms are shown as purple spheres along with a surface map of lipids. I1 is shown as a gray ribbon

colored by domain as in panel (**B**). **E** I1 induces proton flux across liposome membranes. Adding 1 M HCl lowers the pH of the external buffer. When I1 is introduced, the pH gradually increases (pink traces), indicating a leaky membrane. Without I1 (gray traces), the pH remains nearly unchanged until the addition of CCCP to uncouple proton flux. The inset shows a schematic of liposomes used in the assay. Source data are provided as a Source Data file. **F** Calcium influx is measured by fluorescence of Fluo-4 (F4) loaded in SH-SY5Y cells, expressed as a percentage of the maximum capacity determined by cells containing ionomycin, a calcium ionophore. I1 (pink) significantly elevates intracellular calcium levels compared to controls (gray). Error bars represent standard error of the mean obtained with 6 replicates. Inset illustrates $Ca^{2+}$ flux induced by I1. The concentration of αS in I1 used for the proton and $Ca^{2+}$ flux experiments was 0.6 μM. Source data are provided as a Source Data file.

are also variations in the fold of individual filaments. Specifically, the Lewy fold features a near 180° turn at G84, in contrast to L2's approximately 90° turn. Additionally, the turn at G73-is nearly 90° in the Lewy fold and about 160° in L2, leading to a concave bend at G51 in the Lewy fold versus a convex bend at the same position in L2 (Supplementary Fig. 12B).

Despite these differences, the similarities between the Lewy fold and L2 fibrils are striking. Both share the T59-arc (β2-β3) that is common structural kernel in multiple αS fibrils[25]. There are also similarities in other segments, including the interactions between β1 and β4 as well as the organization of β5 (Supplementary Fig. 12C). These similarities make the I1 oligomer from the in vitro L2 fibril preparation a

remarkable model for studying the assembly of structural features present in brain extracted αS fibrils.

The I1 structure is distinct from non-toxic oligomers which are helical and more dynamic such as the dynamic helical tetramer and those stabilized by EGCG (epigallocatechin-3-gallate)[3,26,27]. Additionally, I1 is distinct from stable lipoprotein particles formed by helical αS[28]. While non-toxic oligomers interact with lipid bilayers in an unspecific manner[3,26], the I1 N-terminus contacts lipids via an amphipathic helix and many segments contact the hydrophobic bilayer core. This interaction pattern may influence the cellular fate of αS aggregates because the lipid binding sites also act as recognition motifs for protein quality control, such as for heat

shock proteins (V37-K43 in αS)[29,30] and ubiquitination (K21, K23, K32, K34 in αS)[31].

The I1 model suggests that aggregation prone regions (APRs) are among the first to adopt β-strands in the aggregation process. Identified through factors like hydrophobicity, solubility[32], and mutagenesis studies[33], APRs include the Y39-S42 master controller region, and the K45-E57 segment housing several familial mutations linked to Parkinson's disease[32,34]. In I1 residues in these segments adopt β-strands (β1 and β2 in Fig. 3A) and engage in tertiary and backbone interactions with other APRs (β3 and β4 in Fig. 3A). All of these segments are consistent with those that lend energetic stability to αS fibrils[34]. Additionally, lipid bound dimers reveal helix breaks at V40 and K60[35], suggesting that the destabilization of the functional helix in these segments may steer the molecule toward amyloid aggregation. G84-V95 is a hydrophobic segment; however, instead of a β-strand, it forms a lipid bound loop in I1 (Fig. 4B). This deviation is attributed to its relatively lower aggregation propensity and higher solubility than the other segments[32].

Biophysical commonalities, namely, the ability to disrupt membranes[36] and the presence of anti-parallel β-sheet[8,37–39] have emerged among toxic amyloid oligomers. Consistent with this, the lipid defects caused by I1 (Fig. 4C–E and Supplementary Figs. 11A, B) are reminiscent of edge pores observed with amyloid-β oligomers[40,41] that impair membrane integrity. This impairment has been extensively studied as a mechanism involving increased influx of cations in the context of various amyloid oligomers[36]. The resulting increased influx of cations such as Ca²⁺ is an integral step in the apoptosis signaling pathway which is responsible for the death of dopaminergic neurons[42].

The I1 surface contains more hydrophobic residues, while in the L2-fibril, these are buried in its fold. Previous work[3,29] has shown that αS intermediates are more hydrophobic than fibrils, promoting the absorption of the intermediates exposed hydrophobic surface into the hydrophobic region of lipid bilayers. This is consistent with the I1 structure proposed here. In the absence of lipids, AP β-strands of I1 would have two solvent-exposed interfaces. One interface has a hydrophobic ladder formed by alternating steps of V63 and V55 and the other formed by V66 and V52 (Supplementary Fig. 13A). In addition, residues A69 and V71 are left exposed due to a wider loop at V74 (Supplementary Fig. 13A, bottom). Hydrophobic residues in the N- and C-terminal helices, namely V15, V16, A17, A19 A89, A91, I88 and F94 would also be exposed to the solvent (Supplementary Fig. 13A). The AP to PIR conversion results in the residues V52 and V66 becoming buried in the core of the β-arc formed by two PIR β-strands (Supplementary Fig. 13B), decreasing the solvent exposed hydrophobic surface. Similarly, the V74 loop gets tighter upon the AP to PIR transition, bringing A69 and V71 closer to A78, and reducing their exposure to the solvent (Supplementary Fig. 13B). the hydrophobic residues I88 and F94 remain exposed to the solvent until the C-terminal strand (β5) folds onto the β3 in the L2 fibril, which is after the intermediate 2 stage[17] (Supplementary Fig. 6D).

Anti-parallel β-strands in toxic oligomers were first observed in bulk measurements[8,37] and have since been reported to occur at different residues, featuring varied topologies such as a β-hairpin (L38-A53)[43] or a steric zipper (K80-A91)[38]. Given the polymorphism observed in αS fibrils, multiple intermediate structures can be expected, evidenced by different residues participating in the AP β-strands. However, commonalities in the aggregation pathway are evident, as, similar to I1 oligomers (Supplementary Movie 1), amyloid-β oligomers have also been reported to undergo a 90° turn in a β-hairpin during fibril formation[15]. The presence of β-arcs in most known fibrils suggests a common assembly pathway through a hairpin-to-arc transition.

The conservation of the β-arc structural kernel at T59 in a vast number of αS fibrils (Fig. 1B and Supplementary Fig. 12) suggests that contacts like in the I1 AP domain could be initiators of aggregation for these polymorphs. Tetrameric oligomers modelled from these polymorphs can accommodate I1 type AP β-strands in their otherwise distinct structural folds (Supplementary Fig. 14B) and the formation of such domains appears to be energetically favorable (Supplementary Fig. 15B). This suggests that I1 type AP domains may be at the heart of a common folding pathway for fibril polymorphs with a β-arc at T59 (Supplementary Fig. 15A). The β-arc at G67 is implicated in the formation of MSA-type fibrils (Supplementary Fig. 14A)[25]. Exploring αS oligomers preceding the G67 arc could offer insights into the fibrillogenesis of MSA-type fibrils.

Here we have localized with atomic scale precision, the occurrence of AP β-strands in a toxic αS tetramer. We demonstrate that aggregates containing these AP β-strands precede the formation of fibrillar intermediates, and their transition into a fibril like β-arc is essential for fibril elongation. The observation that the tetramer is in contact with hydrophobic lipid chains and results in detrimental cation influx highlights its potential role in disrupting the integrity of biological membranes, such as those of presynaptic termini, where αS is known to be enriched[44]. Findings here underscore that the tetramer structural models can serve as a basis to investigate genesis, polymorphism and therapeutic intervention for fibrils with similar substructures, like the brain extracted PD/ DLB fibrils. The presence of AP strands may be an early step in triggering the amyloid aggregation process, which has been implicated in various neurodegenerative diseases.

## Methods
### Protein production and tagging with ATTO647N dye
Expression of αS was performed in the *E. coli* strain BL21(DE3). Unlabeled, uniformly ¹⁵N-, uniformly ¹³C- and uniformly ¹⁵N, ¹³C-labeled αS was produced in modified minimal medium. ¹⁵N-NH₄Cl was used as the nitrogen source and ¹³C₆-D-glucose as carbon source, and the protein purification was performed following a published protocol[45]. Cell lysis was achieved by freeze-thaw cycles followed by sonication. The lysed cells were then boiled for 15 min and centrifuged at 48,000 rcf for 45 minutes. Streptomycin (10 mg/ml) was used to precipitate DNA from the supernatant. Following another centrifugation step, ammonium sulfate (0.36 g/ml) was added to the supernatant to precipitate αS. The precipitate was resuspended in 25 mM Tris/HCl, pH 7.7. The protein was further purified by anion exchange chromatography on a POROS HQ column (PerSeptive Biosystems). Mutant αS with an additional C-terminal cysteine (A140C) was generated by PCR-based site-directed mutagenesis (QuikChange 2, Agilent Technologies). The cysteine mutant was tagged with ATTO647N dye (ATTO-TEC) by overnight incubation on ice with a two-fold molar excess of the thiol-reactive maleimide of the dye in phosphate buffered saline (PBS), pH 7.4. Excess dye was removed by gel filtration on a Superdex 75 16/60 HiLoad column (Cytiva). Successful tagging was verified by electrospray ionization mass spectrometry. All protein samples were finally dialyzed against 50 mM HEPES pH 7,4, 100 mM NaCl. The final protein concentration was adjusted to 0.3 mM.

### Intermediate 1 sample preparation for MAS NMR
Aggregation of αS was according to the protocol in Antonschmidt et al.[17]. Monomeric αS in 50 mM HEPES and 100 mM NaCl (pH 7.4) was centrifuged for 1 h at 160,000 rcf (TLA-100.3 rotor in an Optima MAX-TL ultracentrifuge, Beckman Coulter) at 4 °C to remove any large aggregates. The supernatant was decanted and added to a solution of SUVs (small unilamelar vesicles) and NaN₃ (0.02 weight %) to obtain a final protein concentration of 70 μM and a molar Lipid/Protein ratio of 10:1 or 5:1. The mixture was sonicated in cycles for 30 s (20 kHz) with 30 min of quiescence between sonication at 37 °C using a Q700-110 sonication device, with a Microplate Horn Assembly (431MPX) and a Compact Recirculating Chiller (4900-110, all QSonica). The aggregation was monitored with Thioflavin-T fluorescence until the reading

exceeded the background value (~6-8 a.u.) by ~2 units at which point a pellet of I1 was collected via ultracentrifugation. Maintaining temperature at ~4 °C during centrifugation (160,000 rcf) and rotor packing, and ~16 °C for MAS NMR measurements ensured batch-to-batch reproducibility and stability (Supplementary Fig. 1).

ThT fluorescence of the samples was monitored continuously by taking aliquots from the aggregating solution in the sonicator and mixing with a working solution composed of 50 mM glycine buffer at pH 8.5 and 2.5 μM ThT. Measurements were done on the Varian Cary Eclipse fluorescence spectrometer. Fluorescence was excited at 446 nm and emission was recorded from 460 to 560 nm at room temperature. The aggregating solution was vortexed at least once before collecting an aliquot for the ThT measurement. For a single 1.3 mm rotor, about ~ 8 mL of 70 μM αS was aggregated in batches of 1.65 mL. Batch to batch differences in the length of the lag phase were accounted for by measuring ThT fluorescence for each batch independently, and the increase in fluorescence was relative to the initial value of that specific batch. This approach resulted in samples with repeatable spectra (Supplementary Fig. 1D) and the ThT fluorescence measurements for the samples prepared for this study are shown in Supplementary Fig. 1E.

As soon as 2 units increase in fluorescence was detected, samples were placed on ice and centrifuged at 160,000 rcf (TLA-100.3 rotor in an Optima MAX-TL ultracentrifuge, Beckman Coulter) for 1 h at 4 °C. After decanting the supernatant, the pellet was washed with 5 mM HEPES (pH 7.4 and subsequently centrifuged (10 min, 175,000 rcf, 10 °C) twice, each time removing excess moisture. Samples were packed into ssNMR rotors by cutting off the bottom of the centrifuged tube and centrifuging (at 4 °C) the pellet directly into the rotor through a custom-made filling device made from a truncated pipette tip. The rotor was centrifuged in an ultracentrifuge packing device for 30 min at 68,000 rcf in an SW 32 Ti rotor in an Optima L-80 XP Ultracentrifuge (both Beckman Coulter) at 4 °C for packing the pellet[46]. After this step, for 1.3 mm rotors, excess water was pushed out of the rotor by pushing on the bottom rubber seal before closing the rotor. All rotor packing steps were performed in the cold room, as far as possible, using tweezers, to prevent the intermediate from changing states.

## Preparation of SUVs
To produce SUVs, POPC and the sodium salt of POPA, obtained from Avanti Polar Lipids, were dissolved in chloroform and mixed to obtain a 1:1 molar ratio of both lipids. The solvent was evaporated under a nitrogen stream and lyophilized overnight. The lipid film was rehydrated with 50 mM HEPES, pH 7.4 and 100 mM NaCl buffer to a total lipid concentration of 3 mM. The solution was sonicated at 37 kHz for four cycles of 10 min sonication and 10 min rest and filtered through a 0.22 μm syringe filter to obtain SUVs[17].

## ssNMR
All measurements were performed on Intermediate 1 composed of u-[13]C,[15]N-labeled αS. 3D (H)CANH, (HCO)CA(CO)NH, (H)CONH, (H)CO(CA)NH and (HCA)CB(CA)NH experiments[47] for protein sequence assignment and 3D H(H)NH (z-mixing) experiments for lipid-protein contacts were acquired on an 800 MHz Bruker Avance III HD spectrometer at a magnetic field of 18.8 T equipped with a 1.3 mm magic-angle spinning (MAS) HCN probe and MAS at 55 kHz and an estimated sample temperature of 16 °C. The cooling gas flow was set at ~ 1500 liters per hour and temperature of the cooling gas was set to 235 K. The delays for scalar carbon-carbon transfers were set based on the T2' values of 20 ms for Cα and 45 ms for C' as shown in Supplementary Table 2. For backbone assignment experiments, an Intermediate 1 sample aggregated with 10:1 L:P was used. Amine side chains for Q62 and N65 were assigned based on contacts from H(H)NH and corresponding C' assignments from (H)CONH and (H)Ca(CO)NH spectra.

Chemical shifts for Cα, CO, Cβ, $H_N$ and $N_H$ were inputted in TALOS + to obtain predictions on secondary structure and dihedral backbone angles[48]. Secondary structure was confirmed with secondary chemical shift differences between Cα, Cβ resonances and their random coil values calculated according to Schwarzinger et al.[49]. Chemical shift perturbations between fibril and I1 were calculated according to equations in Williamson et al.[50] from [13]C-[13]C and (H)CaNH spectra:

$$\text{Average CSP} = \sqrt{\frac{1}{4}\left[(\delta C\alpha)^2 + (2.37\delta C\beta)^2 + (0.47\delta Co)^2 + (1.30\delta NH)^2\right]}$$
(1)

The H(H)NH pulse sequence used for proton-proton z-mixing measurements is similar to that reported by Najbauer et al. where longitudinal mixing drives proton-proton mixing between the protein and mobile lipid and water molecules[51]. To eliminate spectral overlap between protein side-chain resonances and lipid protons, a $T_2$ filter and a J-filter of 3 ms have been added after the proton excitation pulse.

Partial side-chain assignments were obtained from (H)CCH experiments with [13]C-[13]C RFDR mixing of 1.3 ms. Long range contacts were obtained from H(H)NH and H(H)CH experiments with [1]H-[1]H RFDR mixing of 0.5 ms[52]. These experiments were acquired on a 1200 MHz Bruker Avance NEO spectrometer at a magnetic field of 28.2 T equipped with a 1.3 mm magic-angle spinning (MAS) HCN probe and MAS at 55 kHz and an estimated sample temperature of 16 °C. For these measurements, 100 mM Cu-EDTA was added to the sample in the rotor to an estimated final concentration of 40 mM for sensitivity enhancement[53]. This shortened the recycle delay from 1.6 s to 0.6 s without causing changes to the (h)CaNH spectrum.

The (H)N(H)(H)NH MODIST spectra was acquired on a 1200 MHz Bruker Avance NEO spectrometer at a magnetic field of 28.2 T equipped with a 1.3 mm magic-angle spinning (MAS) HCN probe and MAS of 55,555 Hz with 3.46 ms of [1]H-[1]H MODIST mixing[19]. The sample contained Cu-EDTA at 40 mM. The temperature of the cooling gas was set to 245 K with a flow of 1000 liters per hour. The spectral widths were 40 ppm for [1]H and 38 ppm on [15]N. [1]H and [15]N hard pulses were 100 kHz and 58.8 kHz. The spectra were recorded for 8 days for the 100% labelled sample and 17 days for the 50% labelled sample.

The 2D [13]C-[13]C DARR spectrum with a mixing time of 20 ms was acquired on an 850 MHz Avance III spectrometer with a 3.2 mm MAS HCN probe at a magnetic field of 20.0 T and MAS at 17 kHz.

For all spectra 13.75 kHz MISSISSIPPI water suppression[54] (100 to 200 ms), 12.75 kHz Swf-TPPM proton decoupling during acquisition of the indirect dimension[55] and 10 kHz WALTZ-16 heteronuclear decoupling during acquisition was used.

Spectra were acquired in short blocks of 12-21 hrs for linear drift correction[56]. The drift-corrected blocks were then averaged and processed as one spectrum in Bruker Topspin 3 or 4. Window functions used to process spectra were exponential and quadratic sine. Spectra were analyzed using CcpNmr Analysis.

The stability of the sample during solid-state nuclear magnetic resonance (ssNMR) measurements was monitored by (H)NH spectrum recorded intermittently between blocks of 3D experiment acquisitions. Measurements were halted when the intensity of the spectrum began to reduce, or new peaks appeared in the (H)NH spectrum. A new sample was prepared for further measurements and the reproducibility of the (H)NH spectrum for I1 samples is shown in (Supplementary Fig. 1). I1 samples were remarkably stable (Supplementary Fig. 1) even after 21 days at 55 kHz magic angle spinning (MAS) and an estimated sample temperature of 16 °C. This can be attributed to lack of free monomer available to polymerize I1 into higher order aggregates (Supplementary Fig. 1), relatively slow diffusion in the densely packed pellet in the ssNMR rotor and a sample temperature in the magnet which was much lower than that used for aggregation (37 °C).

Spectra were processed and analyzed on CCPN Analysis 2.4.2[57] and Topspin 4.0.7 (Bruker, AXS GmBH). Signal to noise ratios were determined in Sparky[58].

## Transmission electron microscopy

An I1 sample was resolubilized from a MAS ssNMR rotor and dissolved in 5% glycerol and 5 mM HEPES (pH 7.4) buffer. This was diluted 1:80 times and used to blot TEM grids. Samples were bound to a glow discharged carbon foil covered 400 mesh copper grid. Samples were stained with 1% uranyl acetate aqueous solution and evaluated at room temperature using a TALOS L120C micropscope (Thermo Fisher Scientific). Images were analyzed with ImageJ software[59].

## Center-band only detection of exchange (CODEX) under dynamic nuclear polarization conditions (DNP)

An I1 sample was prepared with $^{13}C$ isotopically labeled site at H50 $^{13}C\epsilon$. This occurs only once in the $\alpha$S sequence. Isolated I1 sample was mixed with TEMTriPol in $^{13}C$-depleted $d_8$-Glycerol, $D_2O$ and $H_2O$ (60:30:10 vol%) to a final 15 mM concentration. This was packed in a 3.2 mm rotor and flash frozen by plunging in liquid nitrogen. The fibril sample was exchanged with glycerol until the mass indicated 60% vol% of glycerol. Then AMUPOL powder was added to a concentration of 30 mM before mixing and plunge freezing the rotor. 395 GHz of microwave irradiation was applied that resulted in a 4 times enhancement for I1 and 30 times enhancement for the fibril sample. All CODEX experiments were measured on a 600 MHz Bruker Avance III HD spectrometer, and a 3.2 mm low temperature (LT) HCN MAS probe at 8 kHz MAS. Using CODEX to count the oligomeric numbers is discussed in detail in previous literatures[20,60,61].

## Photobleaching measurements at room temperature

ATTO647N bound to A140C $\alpha$S was aggregated as outlined in the 'sample preparation' section with wild type (WT) $\alpha$S at a ratio of 1:3 ATTO-$\alpha$S: WT. Once isolated, the sample was packed in a 1.3 mm rotor and a $^{15}N$-$^{1}H$ fingerprint spectrum confirmed that the sample indeed was structurally similar to I1 (Supplementary Fig. 4).

The proteins were diluted to a stock solution of 50 nM in 10 mM HEPES and 10% glycerol at pH 7.8 (working buffer). The protein was further diluted into the working buffer containing 5% poly-vinyl alcohol (PVA) to obtain a final concentration of ~20 pM. Then, 4 µl of this diluted solution was spin-coated onto a plasma-cleaned mirror-enhanced substrate, which was prepared in-house[62]. Finally, the sample was immediately loaded into our custom-built cryogenic microscope[62].

Upon inserting the sample into our custom-built cryogenic microscope, we applied a vacuum for 5 minutes to immobilize the molecules. After releasing the vacuum, we started acquiring videos from multiple fields of view (FOV) at room temperature (RT). The sample was illuminated with a 645 nm wavelength laser at 2 mW in a wide-field (WF) mode using an air objective (Mitutoyo 100X, 0.9 NA). Each FOV (80 × 80 µm) was recorded at a frame rate of 10 Hz for up to 8 min, a time point which shows complete photobleaching in the FOV. Next, we localized and clustered each molecule in the FOV to extract their intensity levels over time. The intensity time traces (Supplementary Fig. 4F and 4G) were then fitted using the DISC algorithm to extract the number of intensity steps per molecule[63]. Here, we used a critical value of 15, and the minimum number of points per cluster was set to 15. To avoid any sources of artifact in the final analysis resulting from low signal-to-noise traces or traces that included high blinking events, we filtered the intensity time traces with a signal-to-noise ratio (SNR) above 5. The output from this analysis (1421 time traces) was then plotted as a histogram and fitted with a binomial model to extract the labeling efficiency or the stoichiometry of the intermediate

oligomers. The binomial distribution described as:

$$P(k) = \binom{n}{k} p^k (1 - p)^{n-k} \qquad (2)$$

In Eq. 2, P is the probability that an oligomer contains k labeled subunits, n is the total number of monomers per oligomer and p is the labeling efficiency. Here we obtain the fit parameter (p), theoretical labeling efficiency, as a function of the total number of monomers (n) as depicted in Supplementary Fig. 4F.

Labeling error is defined as:

$$\left( \frac{|\text{fitted labeling efficiency} - \text{experimental value}|}{\text{experimental value}} \right) \qquad (3)$$

We plotted labeling error (Eq. 3) as a function of monomers per oligomer (red curve) and we found that the best model is a tetramer (black arrow in Supplementary Fig. 4I). Similarly, the residual of the fit indicated that tetramer is the best model (orange curve, Supplementary Fig. 4I).

## Cryogenic polarization measurements (polarCOLD)

The sample was prepared and imaged using the same instrument as in the fluorescence photobleaching experiment. For polarCOLD, the chamber was completely evacuated to a pressure of $1.6 \times 10^{-6}$ mbar and then cooled down to 4.3 K using liquid helium. The setup was then allowed to stabilize for 1-2 h to minimize drift during recordings. Subsequently, the sample was illuminated with a 20 mW laser in WF mode, utilizing the same laser source and microscope objective. The emission signal was split into two channels using a polarized beam splitter, enabling the recording of a polarization time trace[23]. After localizing and clustering each point spread function (PSF), we extracted the polarization time trace, which was then fitted using the DISC algorithm combined with 2D gaussian mixture model of the polarization and coordinate space to determine the number of polarization states per molecule. The number of identified polarization states (dipole orientation) in each PSF corresponds to the number of labeled monomers per oligomer, as the dipole orientation at 4 K of each fluorophores is random but fixed (see Supplementary Fig. 4H). This, in turn, allow us to annotate each fluorophore over time and localize it with high precision beyond the diffraction limit by clustering their coordinates accordingly (Fig. 2E–H). Then, a 2D super-resolved image is reconstructed by assigning a 2D Gaussian function to each localized fluorophore with a width given by the respective localization precision. These super-resolved images demonstrate different projections of the protein molecules within the sample. The output of the number of polarizations (from 1023 traces) was plotted as a histogram (Fig. 2D), yielding results similar to those obtained from the photobleaching steps experiment.

## Determination of $\alpha$S concentration in I1 samples

An I1 sample from an ssNMR rotor, once confirmed to have the expected spectrum, was emptied, resuspended in buffer and aliquots were taken for concentration determination. Aliquots were incubated with 6 M Guanidine Hydrochloride (GdHCl) at room temperature for 2–4 h to dissociate aggregates. Then the sample was loaded onto a 12% SDS-PAGE gel for densitometric analysis and images of the Coomassie stained gel were obtained on a BIORAD Gel Doc XR with Image Lab software. The intensity of the band at ~15 kDa was analyzed with ImageJ to determine the mass of $\alpha$S loaded and converted to concentrations. To correlate intensity of the band with $\alpha$S mass, a standard curve was built where the initial $\alpha$S mass added to the gel was calibrated by measuring absorbance with a 0.2 mm cuvette at 275 nm with an extinction coefficient of 5600 $M^{-1}cm^{-1}$ prior to loading the gel. An attempt was made to measure all I1 samples after GdHCl treatment

with absorbance. However, the presence of a high concentration of lipids often lead to baseline distortions specially in the regions around 180–300 nm. Note that all concentrations are expressed as monomer equivalents.

## Absorbance measurements of αS stocks

Absorbance measurements were performed on an HP Agilent 8453 Diode array spectrophotometer to determine the concentration of αS before using it in aggregation assays with a cuvette of pathlength 0.2 mm. The absorbance was measured at 275 nm and the extinction coefficient was 5600 M$^{-1}$ cm$^{-1}$. For ATTO647N labeled stocks, absorbance was measured at 650 nm with an extinction coefficient of 150000 M$^{-1}$ cm$^{-1}$.

## Determination of amount of lipid in I1 samples

A standard sample of POPC was weighed out and packed in a 1.3 mm rotor and measured at 55 kHz on a 800 MHz spectrometer. The height of the peak at 1.3 ppm of the 1D $^1$H spectrum of this sample was used as the reference to determine the amount of lipid in I1 and L2 samples. A similar 1D $^1$H spectrum was acquired for all subsequent samples and compared to the reference.

## Cell viability assay

SH-SY5Y cells, obtained from ATCC (CRL-2266) were grown in 45% modified eagle media supplemented with L-glutamine (2 mM), HAM's F-12 nutrient mixture (45%), fetal bovine serum (10%) and non-essential amino acids (1%). Cells were grown on Poly-D-lysine coated dishes and one day before treatment, cells were plated in a 96-well plate at a density of $2 \times 10^4$ cells/ well. Samples consisting of the intermediate 1, fibrils or monomers were added to the cells at concentrations defined in Fig. 1 and incubated at 37 °C for 20–24 h. At the end of the treatment period XTT (2,3-bis(2-methoxy-4-nitro-5-sulfophenyl)-5-carboxanilide-2H-tetrazolium) and electron coupling reagent (ThermoFisher Scientific) were added and incubated for another 4 h before reading absorbance at 450 nm and 660 nm. Results are presented after subtraction of blank absorbance at 450 nm and well as background at 660 nm from the test absorbance at 450 nm.

## Calcium flux and cell death assay

SH-SY5Y cells were plated in a 96 well plate at a density of $2 \times 10^4$ cells/ well. The next day, media was aspirated and media containing 3 μM Fluo-4 was added. Cells were incubated at 37 °C for 1 h. Fluo-4 AM containing media was removed and replaced with phenol red free DMEM supplemented with Glutamine and 10% FBS. The plate was incubated for another 10 min to load cells with Fluo-4 and ensure complete de-esterification of Fluo-4 AM. CNQX (cyanquixaline) diluted in media or an equivalent amount of media for control was added to cells to a concentration of 5 μM. Then either I1 in 5 mM HEPES, or an equivalent amount of buffer, monomer, fibrils or ionomycin were added to the cells at a concentration of 0.6 μM. The plate was equilibrated in the BioTEK plate reader at 37 °C for 15 min before measurements began. Fluorescence was excited at 488 nm and measured at 530 nm using a filter cube. In another set of wells, after the overnight incubation, media was replaced with phenol red free media. Either I1 in 5 mM HEPES, or an equivalent amount of buffer, monomer or fibrils were added to the cells at a concentration of 0.6 μM. As a dead cell control, cells were lysed with 10% SDS. Then propidium iodide was added at a concentration of 50 μg/ml. Fluorescence was read in parallel to Fluo-4 with the monochromator based fluorescence module on the BioTEK reader at excitation and emission wavelengths of 535 nm and 622 nm respectively.

## Liposomal proton flux assay

A pH-based proton flux assay was adapted from a previous protocol used for viroporins[64]. Liposomes were made by combining 10 mg of Escherichia coli polar lipid extract (Avanti Polar Lipids) dissolved in chloroform, valinomycin solution in ethanol and methanol in a glass tube. The solvents were evaporated under continuous nitrogen gas and a thin film was obtained. The films were dissolved in chloroform again and dried down under a nitrogen stream and were left overnight in the lyophilizer to remove any solvent trace. The films were then resuspended in strongly buffered internal liposome buffer (26 mM potassium citrate, 17 mM citric acid, 28 mM sodium citrate, 25 mM K$_2$HPO4, 25 mM Na$_2$HPO$_4$, 6 mM NaN$_3$; pH 7.7) to form liposomes which were then extruded 11 times through 0.2 M polycarbonate membrane. Buffer was exchanged on a PD-10 column (GE Health Sciences) such that the external liposome buffer was a weak buffer (4% v/v IVB, 117 mM KCl, 117 mM NaCl, 6 mM NaN$_3$, pH 7.7).

Every tested sample contained 5 mg/mL lipids, and 0.1 μM valinomycin as a potassium ionophore. The external pH was decreased by the addition of 1 M HCl under continuous fast stirring. Once the pH had stabilized, 0.6 μM of I1 in 5 mM HEPES or an equivalent amount of buffer without I1 was added and the pH was recorded every second. The proton uncoupler carbonyl cyanide m-chlorophenylhydrazone (CCCP) was added to determine the buffering capacity of the liposomes.

## CYANA modeling

Contacts from Supplementary Table 1 were used for the CYANA[65] calculation with 250 structures and 80,000 steps. Parallel in-register hydrogen bonds were assumed for segments that show chemical shift similarity ($< 0.7$ ppm average CSP) and high fidelity in long range contacts with the L2 fibril for stretches V31-G51, E57-Q62, and G67-K80. Dihedral angles with good confidence from the TALOS+ prediction were used with one standard deviation with the exception of loop regions, where the limits for the dihedral angles for the loops were increased to three standard deviations. This resulted in the structures with a target function of ~5.

## Molecular dynamic (MD) simulations

To produce an atomistic I1 structure model with restrained MD simulations the core of the α-Synuclein L2 fibril structure (8A4L, residue 33–83) was taken. The N-terminal part was removed. For an N- and C-terminally extended I1 structure model, residues 16–33 & 83–99 were taken from the micelle-bound α-Synuclein monomer structure (1XQ8) and fitted on residue Thr33 & GLu83, respectively. Three different tetrameric I1 AP domain morphologies ('open', bowl' and 'barrel') were derived by MD simulations with distance restraints in a water box (Supplementary Table 1 and Supplementary Fig. 8, Supplementary Fig. 9). In all simulation systems, the titratable amino acids were protonated according to their standard protonation states at pH 7, while also taking into account the solvent exposure and electrostatic interactions with neighboring polar groups. Thus, aspartic and glutamic side-chains were simulated with negative charge and all histidine side-chains were set to neutral. All lysine side-chains were simulated as positively charged[66]. The N- and C-termini of the truncated α-Synuclein molecules were capped with acetyl and N-methyl groups, respectively. All production runs were preceded by a multi-step equilibration of the system. The protein part was separately energy minimized in water. Bilayer patches with a ratio of 1:1 ratio of POPC and POPA lipids and a water slab of 3.5 mm thickness of top and bottom of the bilayer were prepared using the CHARMM-GUI[67] webserver. The membrane patch was relaxed for 1 ns at 300 K. Next, the α-Synuclein structures were either embedded into the lipid bilayer or positioned close to it (see orientations 1/8 in Supplementary Fig. 8) in several different orientations. Subsequently, and if not stated otherwise, Na$^+$ and Cl$^-$ ions (ionic strength: 150 mM) were added in the aqueous phase of the periodic simulation box. Additional MD simulations of I1 in orientation 1 and 2 were carried out with Ca$^{2+}$ ions (salt concentration: 100 mM NaCl and 40 mM CaCl$_2$).

The total simulation size varied and amounted to roughly 65 k or 230 k atoms depending on the used α-Synuclein model short (G36-T81) or long (V16-Q99) construct. Specifications of all simulation systems are summarized in Supplementary Tables 3 and 4. See Supplementary Note 1 for further details.

For the short construct of the open morphology in different orientations, each case was simulated as triplicates of MD simulations with 1000 ns length. For the longer constructs, a total of 62 MD simulations of embedded structures were run for 100 ns with distance restraints and an additional 500 ns without restraints to collect data that are evaluated against experimental measurements (Supplementary Table 4).

The GROMACS 2022 simulation[68,69] software package was used to set up and carry out the MD simulations. Settings for production runs were chosen as follows: The long-range electrostatic interactions were treated using the Particle Mesh Ewald (PME) method[70,71]. Bonds in protein and lipid molecules were constrained using the P-LINCS algorithm[72]. Water molecules were constrained using SETTLE algorithm[73]. Neighbor lists were updated with the Verlet list scheme[69,74]. For production runs, the simulated systems were kept at a temperature of 300 K by applying the velocity-rescaling[75] algorithm. Initial velocities for the production runs were taken according to the Maxwell-Boltzmann distribution at 300 K. The pressure was held constant by using the Parrinello-Rahman barostat[76] with a semi-isotropic coupling in the xy-plane.

All simulations with the CHARMM36m[77,78] protein force field utilized the CHARMM36 lipid[79] parameters together with the CHARMM-modified[80] TIP3P water model. The integration time step was set to 2 fs. The neighbor lists for non-bonded interactions were updated every 20 steps. Real-space electrostatic interactions were truncated at 1.2 nm. The van der Waals interactions were switched off between 1.0 to 1.2 nm and short-range electrostatic interactions were cut-off at 1.2 nm. For pressure coupling the scheme of Parrinello-Rahman[76] was used to hold the system at a pressure of 1 bar (time constant for pressure coupling, $\tau = 5$).

MD simulations were analyzed with the GROMACS 2022 simulation[68,69] software package and post-processed with in-house scripts. For the analysis based on experimental distance restraints and lipid contacts, only the last 250 ns of each simulation trajectory were used to ensure that the results are not biased by the initial equilibration of the simulation system. Data samples were collected every 250 ps. Pairwise interatomic contacts were quantified for every frame using the gmx mindist and gmx hbond programs. Secondary structure analysis was carried out using the DSSP algorithm[81].

The partial densities of water molecules, lipid groups and ions across the simulation box and along the membrane normal direction were computed using the gmx density tool. Histogram binning was done relative to the center of all lipid atoms. Partial density profiles were averaged over all independent trajectory replicates per simulation system. A detailed description is given elsewhere[40]. Structure and MD simulation renderings were produced with Chimera[82] and ChimeraX[83].

## Solution NMR
[15]N-labeled αS was mixed with 50 mM HEPES (pH 7.4) and 100 mM NaCl to obtain samples with 10% $D_2O$ and 100 μM 2,2-dimethyl-2-silapentane-5-sulfonate sodium. Experiments were recorded on a Bruker 700 MHz spectrometer (Avance III HD with CP-TCI HCND probe with z-gradient) at 288 K. [1]H-[15]N-HSQC spectra were acquired using 3-9-19 watergate for water suppression using 256 increments in the indirect dimension and a relaxation delay of 1.2 s. Assignment of the [1]H-[15]N-HSQC spectrum was done by comparison to BMRB entries 16300, 16904, and 18857.

## Reporting summary
Further information on research design is available in the Nature Portfolio Reporting Summary linked to this article.

## Data availability
Assigned chemical shift data for αS Intermediate 1 were deposited in the BMRB under the accession number 52283. Initial coordinate and simulation input files and a coordinate file of the final output are provided through the Edmond data repository at [https://doi.org/10.17617/3.0V1ODV]. NMR spectra are deposited at Edmond under [https://doi.org/10.17617/3.TXND2C]. The PDB structures used for comparison in the article include 8A4L, 8A9L, 7NCA and 7OZG. The L2-fibril chemical shifts were obtained from BMRB accession number 50585. Monomer and C-terminal of I1 was assigned with the help of BMRB accession codes, 16300, 16904, 6968 and 18857. Source data are provided with this paper.

## Code availability
Parameter files are provided through the Edmond data repository at [https://doi.org/10.17617/3.0V1ODV]. The code to identify single-molecules and extract the intensity time traces in the photobleaching experiment was done using custom written MATLAB code. However, any published codes such as ThunderSTORM can be easily used for the purpose, and available from ref. 84. The code to fit the intensity time traces is available from ref. 63. The code to fit the binomial distribution of the number of dyes per aggregate was written in MATLAB and provided as supplementary data 1. The code to analyze the polarCOLD data was written by a previous lab member and described in the following Böning et al.[62]. Code to identify single-molecules and extract the intensity time traces in the photobleaching experiment was done using custom written MATLAB code.

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

## Acknowledgements

We thank M. Wegstroth for the preparation of isotope-labeled α-Synuclein stock solutions and K. Giller for the preparation of dye labeled α-Synuclein. We thank L.Kopecny for taking negative-stain electron micrographs. We thank Mr. J. Schimpfhauser, Mr. J Bienert and Dr. V. N. Belov from the facility for Synthetic Chemistry at the Max Planck Institute for Multidisciplinary Sciences, Göttingen for synthesizing the TEMTriPol-1 radical. This work was supported by the Max Planck Society (to CG, VaS and BLdG) and the Deutsche Forschungsgemeinschaft (DFG, German Research Foundation) under Germany's Excellence Strategy-EXC 2067/1-390729940 (to CG) and the Emmy Noether program to LBA (project number: 397022504).

## Author contributions

Intermediate1 samples were prepared by V.S. and L.A. NMR experiments were performed by V.S., L.A., K.T.M., K.X., E.N. Cell experiments performed by V.S. and M.N. Liposomal proton flux assay was performed by V.S. and M.S. Fluorescence measurements were performed and analyzed by H.M. and F.W. M.D. simulations were performed and analyzed by DM. D.R. acquired electron micrographs. Conceptualization and methodology by B.Ld.G., C.G., L.B.A., V.S. and L.A. S.B. oversaw protein expression and purification. VaS oversaw fluorescence measurements. VS prepared the figures and wrote the initial draft. B.Ld.G., C.G., and L.B.A. supervised the project. All authors contributed to the writing of the manuscript.

## Funding

## Competing interests

The authors declare no competing interests.

## Additional information

[1]NMR Based Structural Biology, Max Planck Institute for Multidisciplinary Sciences, Göttingen, Germany. [2]Department of Theoretical and Computational
Biophysics, Max Planck Institute for Multidisciplinary Sciences, Göttingen, Germany. [3]Max Planck Institute for Science of Light, Erlangen, Germany. [4]Max-
Planck-Zentrum für Physik und Medizin, Erlangen, Germany. [5]Department of Physics, Friedrich-Alexander University of Erlangen-Nürnberg,
Erlangen, Germany. [6]Facility for Electron Microscopy, Max Planck Institute for Multidisciplinary Sciences, Göttingen, Germany. [7]Cluster of Excellence
"Multiscale Bioimaging: From Molecular Machines to Networks of Excitable Cells" (MBExC), University of Göttingen, Göttingen, Germany. [8]Present address:
Brown Laboratory Department of Chemistry and Biochemistry, University of Delaware, Newark, DE, USA. [9]Present address: Center of High Field Imaging,
Nanyang Technological University, Singapore, Singapore. ✉e-mail: bgroot@mpinat.mpg.de; cigr@mpinat.mpg.de; land@mpinat.mpg.de

