## [Transparent Peer Review file · Nature Communications]

Lipidic folding pathway of α -Synuclein via a toxic oligomer

Corresponding Author: Professor Christian Griesinger

Version 0:

Reviewer comments:

Reviewer #1

(Remarks to the Author)

The manuscript from Sant and colleagues reported an atomic-resolution structural characterization of a toxic pre-fibrillar aggregation intermediate (I1) on the pathway to forming lipidic fibrils. This structural reconfiguration occurs in a conserved structural kernel shared by many α S-fibril polymorphs, including extracted fibrils from Parkinson's and Lewy Body Dementia patients. Consistent with reports of anti-parallel β -strands being a defining feature of toxic α S pre-fibrillar intermediates, I1 impacts the viability of neuroblasts and disrupts cell membranes, resulting in an increased calcium influx. Our results integrate anti-parallel β -strands as unique features of toxic oligomers with their significant role in the amyloid fibril assembly pathway. These structural insights have implications for the development of therapies and biomarkers.

The study is interesting, with a large panel of new data. However, I have some comments regarding the biological part of their study.

Fig.1C: There is a lack of information regarding the cellular toxicity assays presented. No dose-response or time-response is provided. SHSY5Y is a limited model; other cellular models, such as primary cultures of dopaminergic and/or cortical neurons, would be a good addition, especially with alpha-synuclein.

Line 118: How was the "0.3 μ M α S" concentration determined? Can the authors explain how they measure the concentration/content of alpha-synuclein aggregates? How do they homogenize/normalize the experiment with different alpha-synuclein concentrations?

Fig.S10E and F (i.e., calcium influx experiments). The authors should add information regarding I1 concentration and other experimental conditions.

Fig.4: The authors describe lipids but do not mention which kind of lipids they refer to. Is there a specificity?

Minor comments:

There are references inserted in the abstract

Reviewer #2

(Remarks to the Author)

The manuscript "Lipidic folding pathway of α -Synuclein via a toxic oligomer" by Sant et al describes a structural investigation of oligomeric species of alpha-synuclein (aSyn) that form on the pathway of amyloid fibrils templated on the surface of acidic lipid vesicles. It is of fundamental importance to characterise the nature of intermediates along the self-assembly of aSyn into amyloids as these are considered the toxic species formed in the context of aSyn aggregation, a process that is intimately connected with the insurgence of synucleinopathies such as Parkinson's disease.

In the quest of achieving a high-resolution understanding of the structural properties of a stable oligomeric intermediate (I1), the authors employed a large number of state-of-the art ssNMR experiments as well as super-resolution microscopy, TEM and other biophysical techniques. The study also characterises the toxicity properties of these aggregates when incubated with neuroblastoma cells.

Of particular note is the present finding that antiparallel (AP) regions co-exist with parallel in register (PIR) regions within the same oligomeric assembly of aSyn. This provides a key model to explain how AP-to-PIR transition may occur when toxic intermediates convert into non-toxic mature fibrils. Thus a better understanding of the energy landscape of mature fibrils is now possible. It is likely that initial AP nucleus is formed (with initial intramolecular beta-hairpins forming at the monomeric level and seeding the self-assembly of a small nucleus), followed by the I1 stabilisation through the mixed AP and PIR regions, and in turn the AP-to-PIR transition of the b2 and b3 described in this paper.

Overall, I believe this is a remarkable work that reached an unprecedented level of structural understanding, covering most of the structured part of the aSyn sequence (residues 1-100) within the I1 tetramer. I have few suggestions/curiosities:

- 1) The experimental evidence clearly indicates that I1 has a mixed topology of AP and PIR, by contrast to the fully PIR L2 fibril, and that both PIR and AP are in contact with the lipids, however, it is still not understood why only I1 (i.e. not L2) disrupts the lipids. Previous works (e.g. refs 9, 11) showed from ANS binding that aSyn intermediates are more hydrophobic than the mature fibrils, likely promoting the absorption of the intermediates' cores into the inner hydrophobic region of the lipid bilayer. The present study, by generating ssNMR informed models of I1 could clarify this aspect by detailing (a) if I1 exposes more hydrophobic residues than L2 and (2) if the local conversion AP-to-PIR reduces the exposure of hydrophobic residues as in a typical protein folding process.
- 2) As in previous lipid-bound aSyn intermediate ssNMR analyses, the INEPT regime detected only the C-terminal residues of I1, indicating that the rest of the protein sequence is sufficiently rigid to be probed in CP spectra. Considering the coverage of the resonances in the first 100 residues, it would be very interesting to probe the backbone dynamics, perhaps with transverse relaxation, of the structured regions of the protein. Are the AP regions more dynamical than PIR?
- 3) I'm puzzled by the lack of resonances assigned in the first 15 residues, while it was possible to assign the segment 16-19 and show that this is in alpha-helical conformation. Assuming that the region 1-15 is also alpha-helical, this should be structured as the segment 16-19, possibly leading to detectable sharp lines. Is there an explanation for this lack of resonances (are peaks perhaps too overlapped to be assigned)?

Additional minor points:

- 4) The contact between K96 and residues around A30 in I1 indicated that the scenario might be different in the A30P PD mutation. Does the modelling suggest possible clues on this mutation?
- 5) The manuscript shows many spectra with 1H-15N correlation but I believe that an additional figure with the 13C-13C DARR 20ms of I1 side-by-side to L2 would give a better clue of the properties of the intermediate species compared with the mature fibril.
- 6) Perhaps the text makes an exaggerate use of amyloid-field jargon. For example the "b-arc" is not a general term in structural biology and its structural topology should be described to the general audience. Similarly, in the abstract it is used "lipidic fibrils", which might generate confusion for the audience not familiar with aSyn aggregation on lipid membranes.

Reviewer #3

(Remarks to the Author)

Sant et al. Provide an elegant analysis of the structural transitions of an alpha-synuclein intermediate and the effect this intermediate has on membrane permeability. The authors use ssNMR to characterize the intermediate and distinguish residue-specific interactions of this state from those of the fibrillar structure. Additionally, the authors characterize the stoichiometry of the oligomeric intermediate and utilize experimental constraints to build potential intermediate state models. Finally, the interactions with lipid vesicles were assessed and used to evaluate MD simulations of the modeled intermediate interacting with a lipid bilayer at different orientations. These models suggested a change in membrane permeability that was then validated experimentally to reveal an effect on Ca²⁺ influx induced by the intermediate not induced by preformed fibrils. The authors provide exciting evidence of the formation of hairpin structures distinct from the beta-arch structure in the endpoint fibril states in vitro and reveal an interesting mechanism for a toxic interaction with membranes.

Questions and Comments

- The authors provide models/curve fits of aggregation assays in Figure 1A and Figure S6C without describing the axis or showing the underlying data resulting in the fit presented.
- Comparison of the beta-hairpin to beta-arch transition between I1 and I2 reveals an exciting, energetic barrier that must be overcome to produce the hallmark fibrillar structures associated with the disease. Can this structural transition be observed with directed simulations initiated from the I1 models to a state that satisfies the PIR constraints observed for I2? Detailing this transition and determining if the interactions with the membrane facilitate the change would provide exciting insights into the structural constraints that dictate the kinetics of fibril formation.
- The method of the MD analysis after discarding the initial 750ns or 250ns of the simulation, the lipid contacts were observed using the remaining 250ns. In Supplementary Note 1, this is described differently, as well as in the figure legend of Fig S8, which states only 100ns were used. Consistency in the method description would clarify how the models were developed. Additionally, showing the stability of the secondary structure in the unrestrained simulations would show a correlation with the experimentally observed state.

- In Figure S3, the contact map shows interactions between residues around position 28 and residue 96. However, these constraints seem less satisfied in the MD models of the long constructs (Fig S9C). In orientation 2, the lipid-protein contacts at the N-terminus and position 99 are also less satisfied throughout the replicate simulations (Fig S9A). Orientation 2 contains the states spanning the membrane and showing the lipid bending. Showing how the N and C terminal interactions change over time would illustrate how the penetration can be driven by the AP domain and not by the termini's more flexible and dynamic interactions.
 - The penetration of Ca²⁺ ions is nicely shown experimentally; however, the Free energy analysis with the addition of Ca²⁺ ions was not described in the methods, details on the simulation length and visualization of intramolecular protein interactions and changes in protein-lipid interactions with the addition of ions would provide a more detailed view of the structural state of I1 as it undergoes this disruptive effect on the membrane.
- Overall, the work is clearly described and illustrates an interesting mechanism for the structural transitions in alpha-synuclein assembly. The presented results and supporting evidence would be of broad interest in the field.

Reviewer #4

(Remarks to the Author)

The authors describe in detail a new intermediate aggregate of alpha synuclein that is on pathway to the formation of lipidic fibrils. They use a plethora of experimental techniques to describe in detail the structure of this intermediate structure and how it transitions into a fully formed fibrillar structure.

The manuscript is well written and of interest to the readership.

I only have a few points that need clarification.

1. First, a test was performed for cell viability whereas the authors should have used an Anova to measure significance.
2. The labelling of alpha synuclein was not described in enough detail. Was maleimide labelling used, if yes, please state. Also, how was the labelling ratio determined, in line 513 of the manuscript they state that they used MS (no data shown-please show!), whereas in the supplementary Fig. 4b they say that they measured it using nanodrop, which is highly unprecise, especially if the sample is then used to determine whether the intermediate is a tetramer or not.
3. Related to the above, it is not clear to me how the bleaching experiments are performed with a labelling ratio of 1:4. The same goes for the CODEX assay.
4. Also, the effect of the dye may not have a large effect when one compares NMR spectra of labelled and unlabelled protein, but it has a strong effect on the aggregation dynamics, which has not been addressed in this study.
5. Please provide some brief description of how the CODEX assay works rather than referring to other publications.
6. ThT data shown are highly rudimentary, if included show details of traces and reproducibility by including the standard deviation. Also, in suppl Fig. S1 E is not referred to in the legend.
7. On page 3, line 91-93. "The I1 sample can be isolated for prolonged times in the rotor because it is depleted of disordered monomer and membrane bound monomer (Fig.1A)." Not quite sure what this means. From the context it is supposed to demonstrate how they separated the intermediate from the monomer, but it is not clear how this was done.
8. For Figure 1F, the way the authors labelled the distance between side chains doesn't seem to be following a particular rule (for example from C_{beta} - C_{beta} from one side chain to the other). Please demonstrate it more clearly.

Version 1:

Reviewer comments:

Reviewer #1

(Remarks to the Author)

All my concerns were satisfactorily addressed by the authors.

Reviewer #2

(Remarks to the Author)

The revised version of the article "Lipidic folding pathway of α -Synuclein via a toxic oligomer" addresses my previous points.

Reviewer #3

(Remarks to the Author)

The authors of Sant. et al. carefully addressed all reviewer comments in the manuscript and improved the rigor of the exciting results presented. The additional MD simulations with the extra analysis and the improved description of the methods improve the interpretation of the atomic detail provided in the models. The authors implemented a range of updates to the simulations that provide a fuller picture of how the transitions observed in the model are driven by interactions

observed in vitro. Additionally, the data interpreting the stability of the truncated I2 oligomer at different sizes and with the lipids shows interesting points to investigate further in other future studies. The authors have presented an improved manuscript with a detailed description of the mechanistic steps transitioning between structural states for alpha-synuclein. This work describes a novel state analysis with supporting experimental and computational models that will be of broad interest in the field.

Reviewer #4

(Remarks to the Author)

I am happy with the refined version of the manuscript

REVIEWER COMMENTS

Reviewer #1 (Remarks to the Author):

The manuscript from Sant and colleagues reported an atomic-resolution structural characterization of a toxic pre-fibrillar aggregation intermediate (I1) on the pathway to forming lipidic fibrils. This structural reconfiguration occurs in a conserved structural kernel shared by many α S-fibril polymorphs, including extracted fibrils from Parkinson's and Lewy Body Dementia patients. Consistent with reports of anti-parallel β -strands being a defining feature of toxic α S pre-fibrillar intermediates, I1 impacts the viability of neuroblasts and disrupts cell membranes, resulting in an increased calcium influx. Our results integrate anti-parallel β -strands as unique features of toxic oligomers with their significant role in the amyloid fibril assembly pathway. These structural insights have implications for the development of therapies and biomarkers.

The study is interesting, with a large panel of new data. However, I have some comments regarding the biological part of their study.

Fig.1C: There is a lack of information regarding the cellular toxicity assays presented. No dose-response or time-response is provided. SHSY5Y is a limited model; other cellular models, such as primary cultures of dopaminergic and/or cortical neurons, would be a good addition, especially with alpha-synuclein.

Cell viability data for a higher concentration of I1 is included in Fig.1C. At 0.6 μ M α S in I1, the viability decreases compared to 0.3 μ M α S.

For time-response measurements, we refer the reviewer to Fig.4F and Fig.S11. The Calcium influx assay (Fig.4F) and Propidium Iodide fluorescence (PI) (Fig.S11A) are time-resolved measurements. The Calcium influx assay shows that within 15 minutes, the Calcium influx in I1-treated cells exceeds that of control cells, and this difference continues to grow up to 110 minutes. At 110 minutes, the Fluo-4 fluorescence, which increases in response to Calcium concentrations, reaches the maximum achieved by ionomycin (a calcium ionophore), indicating complete disruption of the cell membranes (Fig.4F). Complementarily, the bulk measurement of cell death, observed through the increase in PI fluorescence of the cell-permeable dye (Fig.S11A), shows that cell viability drops around 115 minutes after the addition of I1 and continues to decrease even 24 hours after.

Regarding the use of additional cellular models, we emphasize that the primary focus of this manuscript is structural characterization. The cell viability experiments serve to complement the structural findings with functional assays. SH-SY5Y cells are a well-established model in this context. While we acknowledge the reviewer's suggestion for using primary dopaminergic or cortical neurons, a more detailed biological characterization of I1's effects is beyond the scope of this study. We appreciate the suggestion and will consider it for future research.

Line 118: How was the "0.3 μ M α S" concentration determined? Can the authors explain how they measure the concentration/content of alpha-synuclein aggregates? How do they homogenize/normalize the experiment with different alpha-synuclein concentrations?

We thank the reviewer for pointing out that this was missing in the Methods section. A section called "Determining α S concentration in I1 samples" has been added to the Methods. Briefly, an I1 sample from an ssNMR rotor, once confirmed to have the expected spectrum, was emptied, resuspended in buffer and aliquots were taken for concentration determination.

Aliquots were incubated with 6M Guanidine Hydrochloride (GdHCl) at room temperature for 2-4 hrs to dissociate aggregates. Then the sample was loaded onto a 12% SDS-PAGE gel for densitometric analysis and images of the Coomassie stained gel were obtained on a BIORAD Gel Doc XR with Image Lab software. The intensity of the band at ~15kDa was analyzed with ImageJ to determine the mass of α S loaded and converted to concentrations. To correlate intensity of the band with α S mass, a standard curve was built where the initial α S mass added to the gel was calibrated by measuring absorbance with a 0.2 mm cuvette at 275 nm with an extinction coefficient of $5600 \text{ M}^{-1} \text{ cm}^{-1}$ prior to loading the gel. An attempt was made to measure all I1 samples after GdHCl treatment with absorbance. However, the presence of a high concentration of lipids often lead to baseline distortions specially in the regions around 180 – 300 nm. Note that all concentrations are expressed as monomer equivalents.

Fig.S10E and F (i.e., calcium influx experiments). The authors should add information regarding I1 concentration and other experimental conditions.

The requested information has been added in the new figure Fig.S11.

Fig.4: The authors describe lipids but do not mention which kind of lipids they refer to. Is there a specificity?

A 1:1 mixture of POPC and POPA was the lipid composition for all samples used throughout the study, and the bilayer composition in the simulations. Other lipid compositions were not tested. This information has been added to the legend in Fig4 and in Line 74.

Minor comments:

There are references inserted in the abstract.

These have now been removed.

Reviewer #2 (Remarks to the Author):

The manuscript “Lipidic folding pathway of α -Synuclein via a toxic oligomer” by Sant et al describes a structural investigation of oligomeric species of alpha-synuclein (aSyn) that form on the pathway of amyloid fibrils templated on the surface of acidic lipid vesicles. It is of fundamental importance to characterise the nature of intermediates along the self-assembly of aSyn into amyloids as these are considered the toxic species formed in the context of aSyn aggregation, a process that is intimately connected with the insurgence of synucleinopathies such as Parkinson’s disease.

In the quest of achieving a high-resolution understanding of the structural properties of a stable oligomeric intermediate (I1), the authors employed a large number of state-of-the art ssNMR experiments as well as super-resolution microscopy, TEM and other biophysical techniques. The study also characterises the toxicity properties of these aggregates when incubated with neuroblastoma cells.

Of particular note is the present finding that antiparallel (AP) regions co-exist with parallel in register (PIR) regions within the same oligomeric assembly of aSyn. This provides a key model to explain how AP-to-PIR transition may occur when toxic intermediates convert into non-toxic mature fibrils. Thus a better understanding of the energy landscape of mature fibrils is now possible. It is likely that initial AP nucleus is formed (with initial intramolecular beta-hairpins forming at the monomeric level and seeding the self-assembly of a small nucleus), followed by the I1 stabilisation through the mixed AP and PIR regions, and in turn the AP-to-PIR transition of the b2 and b3 described in this paper.

Overall, I believe this is a remarkable work that reached an unprecedented level of structural understanding, covering most of the structured part of the aSyn sequence (residues 1-100) within the I1 tetramer.

I have few suggestions/curiosities:

1) The experimental evidence clearly indicates that I1 has a mixed topology of AP and PIR, by contrast to the fully PIR L2 fibril, and that both PIR and AP are in contact with the lipids, however, it is still not understood why only I1 (i.e. not L2) disrupts the lipids. Previous works (e.g. refs 9, 11) showed from ANS binding that aSyn intermediates are more hydrophobic than the mature fibrils, likely promoting the absorption of the intermediates' cores into the inner hydrophobic region of the lipid bilayer. The present study, by generating ssNMR informed models of I1 could clarify this aspect by detailing (a) if I1 exposes more hydrophobic residues than L2 and (2) if the local conversion AP-to-PIR reduces the exposure of hydrophobic residues as in a typical protein folding process.

Yes, I1 exposes more hydrophobic residues than the L2-fibril as shown in Fig.S13. The following discussion has been added to the manuscript (lines 382-399):

The I1 surface contains more hydrophobic residues, while in the L2-fibril, these are buried in its fold. Previous work^{3, 28} has shown that α S intermediates are more hydrophobic than fibrils, promoting the absorption of the intermediates exposed hydrophobic surface into the hydrophobic region of lipid bilayers. This is consistent with the I1 structure proposed here. In the absence of lipids, AP β -strands of I1 would have two solvent-exposed interfaces. One interface has a hydrophobic ladder formed by alternating steps of V63 and V55 and the other formed by V66 and V52 (Fig.S13A). In addition, residues A69 and V71 are left exposed due to a wider loop at V74 (Fig.S13A, bottom). Hydrophobic residues in the N- and C-terminal helices, namely V15, V16, A17, A19 A89, A91, I88 and F94 would also be exposed to the solvent (Fig.S13A). The AP to PIR conversion results in the residues V52 and V66 becoming buried in the core of the β -arc formed by two PIR β -strands (Fig.S13B), decreasing the solvent exposed hydrophobic surface. Similarly, the V74 loop gets tighter upon the AP to PIR transition, bringing A69 and V71 closer to A78, and reducing their exposure to the solvent (Fig.S13B). However, the hydrophobic residues I88 and F94 remain exposed to the solvent until the C-terminal strand (β 5) folds onto the β 3 in the L2 fibril, which is after the intermediate 2 stage¹⁷ (Fig.S6D).

2) As in previous lipid-bound aSyn intermediate ssNMR analyses, the INEPT regime detected only the C-terminal residues of I1, indicating that the rest of the protein sequence is sufficiently rigid to be probed in CP spectra. Considering the coverage of the resonances in the first 100 residues, it would be very interesting to probe the backbone dynamics, perhaps with transverse relaxation, of the structured regions of the protein. Are the AP regions more dynamical than PIR?

We thank the reviewer for the interesting suggestion. An (H)NH spectrum can be obtained in a reasonable amount of time, however, only a few resonances are resolved. Therefore, T1 ρ measurements would need to be acquired with an (H)CANH or (H)CONH spectrum, given that a better dispersion of resonances allows us to resolve more peaks. Obtaining a signal to noise ratio of around 30 on an (H)CANH would take 9 days. Thus, multiple 9-day measurements (for each spin lock time) would be needed or around two months to collect enough points to ensure a decay is observed. The limitations on sensitivity and the transient nature of I1, make these measurements extremely challenging but we will keep the suggestion in mind for future work.

3) I'm puzzled by the lack of resonances assigned in the first 15 residues, while it was possible to assign the segment 16-19 and show that this is in alpha-helical conformation. Assuming that the region 1-15 is also alpha-helical, this should be structured as the segment 16-19, possibly leading to detectable sharp lines. Is there an explanation for this lack or resonances (are peaks perhaps too overlapped to be assigned)?

This is indeed curious, and we thank the reviewer for bringing it to our attention. There are two considerations here. First, the affinity of the first ~10 residues of α S for lipid bilayers is dependent on N-terminal acetylation (Maltsev et.al. *Biochemistry* 2012, 51, 25, 5004–5013, Kang et.al. *Protein Science* 2012, 21, 7, 911-917). Previous works have shown that the helical propensity of the first ~10 residues increases upon N-terminal acetylation. In this study, we use nonacetylated α S, suggesting that there may be a smaller population of residues 1-15, than 16-22, that are bound to the lipid membrane and helical.

The second consideration is that residues 10KAKEG14 are one of the many imperfect KTKEGV repeats in the α S sequence. Some of these repeats are part of helices (21KTKQG25) and many are part of loops (32-36, 43-47, 57-61). Chemical shifts in loops appear to show much more dispersion than in helices, suggesting that overlap could indeed be a reason for not observing the first 15 residues. This is also reflected in how only tentative assignments were possible for K23 and Q24 because of overlap in CO resonances and 25GV26 remain unassigned.

Additional minor points:

4) The contact between K96 and residues around A30 in I1 indicated that the scenario might be different in the A30P PD mutation. Does the modelling suggest possible clues on this mutation?

Unfortunately, the I1 structure does not provide any direct clues that might explain the effect of the A30P mutation. There is no reported fibril structure for A30P, and the mechanism for A30P toxicity is generally understood to involve long-lived oligomers (inhibition of fibrils). However, it is not clear how this understanding can be linked to the I1 structure. In the I1 structure, A30 occurs in a loop, buried in the lipid bilayer, connecting the N-terminal helix and PIR β -strands. Proline is generally considered a helix breaker and is frequently found in loops. Given the model, a Proline mutation at this position would not majorly perturb the structure, since the helix is already broken at this position in I1. Furthermore, A30 is followed by G31, which also supports the formation of a loop.

It is entirely conceivable that the A30P mutant drives aggregation toward a different fibril fold than the L2-fibril, implying that the oligomer leading up to such a fibril might be distinct from I1 as well. Since there is no reported fibril structure of an A30P synuclein, it is thus very difficult to predict the effects of the mutation. Even for the L2-fibril A30 is outside the structured region.

5) The manuscript shows many spectra with 1H-15N correlation but I believe that an additional figure with the 13C-13C DARR 20ms of I1 side-by-side to L2 would give a better clue of the properties of the intermediate species compared with the mature fibril.

The ¹³C-¹³C DARR spectra for I1 and the L2-fibril have previously been analyzed in detail in Antonschmidt et.al. *Science Advances* 2021. We have also included the spectra along with assignments in Fig.S1B.

6) Perhaps the text makes an exaggerate use of amyloid-field jargon. For example the “b-arc” is not a general term in structural biology and its structural topology should be described to the general audience. Similarly, in the abstract it is used “lipidic fibrils”, which might generate confusion for the audience not familiar with aSyn aggregation on lipid membranes.

To address this, we have included a schematic to illustrate structural features of a β -arc (Fig.S6A) and lines 247-252. We have also specified what is meant by “lipidic fibrils” in the abstract: “...which incorporate lipid molecules on protofilament surfaces during fibril growth on membranes.”

Reviewer #3 (Remarks to the Author):

Sant et al. Provide an elegant analysis of the structural transitions of an alpha-synuclein intermediate and the effect this intermediate has on membrane permeability. The authors use ssNMR to characterize the intermediate and distinguish residue-specific interactions of this state from those of the fibrillar structure. Additionally, the authors characterize the stoichiometry of the oligomeric intermediate and utilize experimental constraints to build potential intermediate state models. Finally, the interactions with lipid vesicles were assessed and used to evaluate MD simulations of the modeled intermediate interacting with a lipid bilayer at different orientations. These models suggested a change in membrane permeability that was then validated experimentally to reveal an effect on Ca²⁺ influx induced by the intermediate not induced by preformed fibrils. The authors provide exciting evidence of the formation of hairpin structures distinct from the beta-arch structure in the endpoint fibril states in vitro and reveal an interesting mechanism for a toxic interaction with membranes.

Questions and Comments

- The authors provide models/curve fits of aggregation assays in Figure 1A and Figure S6C without describing the axis or showing the underlying data resulting in the fit presented

We would like to point out that Fig1A is just a schematic to illustrate the goals of the study. ThT curves along with fits, raw data and standard deviation from four repeats have been included in Fig.S1E and reproduced in Fig.S6C.

- Comparison of the beta-hairpin to beta-arch transition between I1 and I2 reveals an exciting, energetic barrier that must be overcome to produce the hallmark fibrillar structures associated with the disease. Can this structural transition be observed with directed simulations initiated from the I1 models to a state that satisfies the PIR constraints observed for I2? Detailing this transition and determining if the interactions with the membrane facilitate the change would provide exciting insights into the structural constraints that dictate the kinetics of fibril formation.

The transition from I1 to fibrils occurs via I2 (Antonschmidt *et al. Sci Adv* 7, eabg2174 2021, Fig.S6). While certain aspects are known about I2, like the AP to PIR transition and that the β 5 is not yet folded, experimentally, we still have many unknowns. For example, is this transition driven by the addition of monomers to I1 or is it just kinetically limited? We are also unclear about the driving force behind the β 5 folding on β 3. Efforts to answer these questions are in progress. Unfortunately, ‘brute force’ simulations of the I1 to I2 transition or the de novo assembly from free α Syn monomers is prohibitive due to the long time scales involved in the process, that are not yet accessible even with state-of-the-art MD simulations for the systems in question. Using directed or targeted MD simulations between the two states I1 and I2, however will arguably face the challenge of accurately sampling all relevant configurations of the transition. Running additional MD simulations utilizing truncated models (G36-T81), we have looked at the stability of the L2 fold in the PIR domain and the

AP domain as a function of aggregate size and environment. Indications are that both the successive addition of monomers, as well as the presence of lipid molecules stabilized the L2 fold in I2 oligomers (see **Fig R1**). Although these observations are interesting, they warrant further investigation to carefully examine and separate the exact sequence and specifics of the individual events relevant to the I1 to I2 transition (β -arc folding, loss of lipid contacts in the AP domain, oligomer growth).

Fig. R1. Size-dependent stability of L2 fold in I2 oligomers. RMSD (Mainchain + C β atoms) for truncated models (G36-T81) of the monomeric L2 fibril conformation as function of aggregate size i.e. number of stacked α S molecules in L2 conformation for residues H50-G67 and the PIR domain (res. 37-44 & 75-80). Circles indicate the average RMSD sampled from the last 50ns of three 250ns long simulations in water and salt without membrane (colors: Monomer – light-gray to Heptamer – dark-gray) and membrane-inserted (Tetramer in Orientation1 – light-green, Tetramer in Orientation 2 – dark-green).

- The method of the MD analysis after discarding the initial 750ns or 250ns of the simulation, the lipid contacts were observed using the remaining 250ns. In Supplementary Note 1, this is described differently, as well as in the figure legend of Fig S8, which states only 100ns were used. Consistency in the method description would clarify how the models were developed. Additionally, showing the stability of the secondary structure in the unrestrained simulations would show a correlation with the experimentally observed state.

We thank the reviewer for pointing out the inconsistency and for giving us the opportunity to clarify the explanation of the simulation analysis. Irrespective of the overall length of the trajectories only the last 250 ns were used throughout all analyses. We corrected the text in the Methods section and Figure captions. We also added Tables S3 and S4 to the Supplementary Material for a more detailed overview of the (length of the) MD simulations. Plots of a secondary structure content analysis were included in a new Fig.S10 in the Supplementary

Material that show a) the stability and the chosen I1 simulation models in terms of secondary structure elements as defined by the DSSP algorithm and b) the high correlation with the experimental assignment.

- In Figure S3, the contact map shows interactions between residues around position 28 and residue 96. However, these constraints seem less satisfied in the MD models of the long constructs (Fig S9C). In orientation 2, the lipid-protein contacts at the N-terminus and position 99 are also less satisfied throughout the replicate simulations (Fig S9A). Orientation 2 contains the states spanning the membrane and showing the lipid bending. Showing how the N and C terminal interactions change over time would illustrate how the penetrance can be driven by the AP domain and not by the termini's more flexible and dynamic interactions.

We would like to clarify that both Orientation 1 and 2 span the membrane and show lipid bending and have a disruptive effect on the membrane (Orientation 1, free energy plot Fig.S10B left, and Orientation 2- right). In the previous simulations, the N- and C- termini are quite far away from each other in the starting coordinates of the MD simulations (see Fig R2, left panels).

The protocol to obtain the atomistic I1 structure models was carried out as follows: Starting from the L2-fibril structure, with the termini taken from micelle-bound α -Synuclein monomer structure, the AP β -strands were formed by imposing distance restraints. These were later removed after equilibration and unrestrained simulations were run. However, no such distance restraints were imposed for the contacts between E28/A30-K96. Instead, this contact was used for validation of the structure and we previously showed that its spontaneous formation is an indication of a model close to experimental contacts. N- and C-terminal interactions were indeed established, however, not in all simulation and only after hundreds of nanoseconds of sampling (see **Fig R2, left panels**).

We undertook a considerable effort to supplement and update the current study with MD simulations that explicitly included the 28/30-96 contacts by initially imposing these distance restraints in the starting models (see Tables S3 and S4). All other aspects of the simulation protocol were left unchanged. As one can see, these new/updated MD simulations do satisfy close N- to C-terminal distances more often in such a set-up (see **Fig R2, right panels**). We used these sets of MD simulations now also for the distance restraints, secondary structure and permeability analysis (see Fig. S10).

Fig. R2. N- to C-terminal distances in I1 structure models. Exemplary time-course analysis of the HA-HA E28 to K96 and HN-HN A30 to K96 distances for MD simulations of I1 models (open AP morphology; construct: V16-Q99) in Orientation 1 without (left) and with (right) additional structure restraints to improve the N- to C-terminal contacts as observed in experiments.

- The penetrance of Ca²⁺ ions is nicely shown experimentally; however, the Free energy analysis with the addition of Ca²⁺ ions was not described in the methods, details on the simulation length and visualization of intramolecular protein interactions and changes in protein-lipid interactions with the addition of ions would provide a more detailed view of the structural state of I1 as it undergoes this disruptive effect on the membrane.

We thank the Reviewer for pointing this out. We have expanded and streamlined the description of the methods used to analyze the MD simulations. Details of the protein-protein distance restraints, lipid contacts, secondary structure and permeability analysis for the MD simulations with Ca²⁺ ions have been included in a new Fig. S10. We would like to clarify that (permeating)

ions don't alter the structure of the protein or specific, lipid-inserted regions during the simulations. Rather, they are conducted due to the impact of the protein structure on the membrane integrity, i. e. creating polar defects in the hydrophobic bilayer center. As judged from the hundreds of nanoseconds long MD simulations, apart from the initial equilibration of the membrane inserted oligomers no significant changes in protein secondary structure elements specifically in the AP domain were found over time and across all probed simulation systems (see Fig. S10).

Overall, the work is clearly described and illustrates an interesting mechanism for the structural transitions in alpha-synuclein assembly. The presented results and supporting evidence would be of broad interest in the field.

Reviewer #4 (Remarks to the Author):

The authors describe in detail a new intermediate aggregate of alpha synuclein that is on pathway to the formation of lipidic fibrils. They use a plethora of experimental techniques to describe in detail the structure of this intermediate structure and how it transitions into a fully formed fibrillar structure.

The manuscript is well written and of interest to the readership.

I only have a few points that need clarification.

1. First, a test was performed for cell viability whereas the authors should have used an Anova to measure significance.

We thank the reviewer for pointing this out. Statistical analysis has been redone with a one-way ANOVA Tukey test and all significant comparisons are shown in Fig.1C.

2. The labelling of alpha synuclein was not described in enough detail. Was maleimide labelling used, if yes, please state. Also, how was the labelling ratio determined, in line 513 of the manuscript they state that they used MS (no data shown-please show!), whereas in the supplementary Fig. 4b they say that they measured it using nanodrop, which is highly unprecise, especially if the sample is then used to determine whether the intermediate is a tetramer or not.

We would like to bring to the reviewer's attention that mass spectrometry was used at the stage after expression of the protein and binding of the dye to determine if all of the protein is bound to dye in the stock. The requested Mass spectrometry data has been included in Fig.S4B. Subsequently, an aliquot of this dye labeled stock was mixed with dye-unlabeled stock. At this stage, to determine the percentage of dye labeled protein in this mixture, absorbance measurements were used. We would like to highlight that *no nanodrop was used in the study*. Instead, a cuvette of pathlength 0.2 mm was used to load the samples on an HP Agilent spectrophotometer. A section has been included in the Methods called "Absorbance measurements of α S stocks" to outline this.

3. Related to the above, it is not clear to me how the bleaching experiments are performed with a labelling ratio of 1:4. The same goes for the CODEX assay.

We would like to clarify that the CODEX assay is performed with I1 that was prepared with α S containing a single ^{13}C isotopically labeled site at H50 C ϵ . In this case, every molecule in

each aggregate would have an H50 Cε that is isotopically labeled and there is no dilution of the label.

On the other hand, photobleaching was performed with I1 prepared with a mixture of wild type αS and A140C αS with ATTO647N at a ratio of 3:1. In this scenario, assuming stochastic mixing, we can expect a distribution of aggregates with varying number of dye-labeled molecules. Some aggregates may contain all dye-labeled molecules, some may have none and other may have intermediate ratios (e.g. 75% or 50% labeled molecules). Each dye molecule photobleaches in discrete steps, and the distribution of the dye labels in aggregates, and the associated probability of observing a certain number of photobleaching steps follows a binomial distribution.

This has been clarified under the section “Oligomer state of I1”, line 191-223.

4. Also, the effect of the dye may not have a large effect when one compares NMR spectra of labelled and unlabelled protein, but it has a strong effect on the aggregation dynamics, which has not been addressed in this study.

We would like to clarify that a mutation and dye tagging, specifically at the A140C position does not affect aggregation kinetics. This is shown in Fig.S4E. We attribute this to the location of the mutation in the disordered domain, far away from the structured portion of I1. We agree with the reviewer that mutations and dye tagging in the structured domain may have an impact on aggregation kinetics and structure. We indeed tried to label the protein in the structured region but the aggregation kinetics and more importantly, the spectra looked different indicating that other structures are formed.

5. Please provide some brief description of how the CODEX assay works rather than referring to other publications.

A more detailed description of the CODEX measurement is provided in lines 195-205.

“An NMR CODEX²¹ (Center band only detection of exchange) measurement allows for spin counting with an upper distance limit of about 10 Å. When each molecule is labeled at a single site, CODEX can be used to determine the oligomer number, provided that the labeled sites form a cluster with the nearest intra-spin distance below 10 Å. For these measurements, I1 was prepared with αS containing a single ¹³C isotopically labeled site at H50 Cε. A CODEX measurement involves the decay of initial magnetization of this single isotope labeled nucleus until the signal plateaus at the inverse of the number of spins over which magnetization equilibrates. The CODEX curve reaches about 0.25 at long times, indicating that I1 is at least a 4-mer (Fig.S4A).”

6. ThT data shown are highly rudimentary, if included show details of traces and reproducibility by including the standard deviation. Also, in suppl Fig. SI E is not referred to in the legend.

We thank the reviewer for pointing out this error in the figure legend. An additional panel has been added to Fig.S1 that shows the standard deviation of aggregation kinetics determined by ThT fluorescence as well as the fit, and raw data points to show the aggregation curves of I1 samples used. The kinetics parameters have also been reported in detail in Antonschmidt et.al. Science Advance 2021 (reference 17).

7. On page 3, line 91-93. "The I1 sample can be isolated for prolonged times in the rotor because it is depleted of disordered monomer and membrane bound monomer (Fig.1A)." Not quite sure what this means. From the context it is supposed to demonstrate how they separated the intermediate from the monomer, but it is not clear how this was done.

Thanks for pointing out this confusing/ speculative statement, which has now been improved. The Methods section describes in detail how I1 is separated from monomers by ultracentrifugation. This is also better integrated in the main texts, lines 92-96 of the manuscript:

The I1 sample can be isolated for prolonged times in the rotor (several weeks), which we attribute to a reduction in temperature from 37°C during aggregation to below about 20°C during NMR measurements. Additionally, stability might be improved because I1 has been depleted in disordered monomer and membrane bound monomer via ultracentrifugation before packing.

8. For Figure 1F, the way the authors labelled the distance between side chains doesn't seem to be following a particular rule (for example from C β - C β from one side chain to the other). Please demonstrate it more clearly.

The spectra that were recorded for these measurements ((H)CHH and (H)NHH) allowed for the determination of atom resolved contacts. Therefore, the H α of one residue can be resolved from the H β and so on. The side-chain contacts are labeled according to the resonances observed as explained in lines 151-156. In short, these are atom-resolved contacts.